# Ultralight soft electrostatic actuators based on solid-liquid-gas architectures

Hyeong-Joon Joo [1], Toshihiko Fukushima [1], Xiying Li[1], Alona Shagan Shomron [1], Soo Jin Adrian Koh [1], Philipp Rothemund [1,2] ✉ & Christoph Keplinger [1,3,4] ✉

Soft actuators enable versatile and adaptable robots capable of operating in unstructured environments and close to humans. Soft electrostatic actuators utilizing electrohydraulic principles are particularly promising, combining all-around actuation performance with portable driving electronics. These electrohydraulic actuators harness liquid dielectrics enclosed in solid dielectric shells to sustain high electric fields; the liquid dielectric however constitutes most of the actuator mass, limiting power-to-weight ratio. Here, we present ultralight soft electrostatic actuators based on solid-liquid-gas architectures: the introduction of gaseous dielectrics as a third phase substantially improves power-to-weight ratio by reducing actuator mass and increasing actuation speed. Through theoretical and experimental analyses, we pinpoint the fundamental performance limit as the electrical breakdown in the gas, governed by Paschen's law, thereby providing a guideline for selection of gaseous dielectrics. Using the Peano-HASEL (hydraulically amplified self-healing electrostatic) actuator as a model system, we identify a gas mixture of $C_4F_7N$ and $CO_2$ that enables outstanding specific energy of 51.4 J kg$^{-1}$ (a nine-fold improvement over conventional Peano-HASELs); using ambient air as gaseous dielectric we still achieve 33.5 J kg$^{-1}$ and a power-to-weight ratio of 1600 W kg$^{-1}$ (a five- and eleven-fold improvement). We illustrate these enhanced performance metrics in a jumping robot, showing a 60% increase in jump height, highlighting the wide potential of ultralight soft electrostatic actuators for adaptable and agile robotic systems.

Robotic systems greatly impact our society by performing tasks with superhuman precision, strength, and repeatability, particularly in controlled environments such as factory floors[1]. However, conventional robots that are based on rigid materials often struggle in situations that require versatility and adaptability, and they can be unsafe near people[2–4]. In contrast, soft robots are based on deformable materials, inspired by structures found in nature such as muscles and skin[5], and they show promise for operation in unstructured environments and near the human body[3,4,6–9]. To drive soft robotic systems, a variety of actuation technologies have been explored[10,11], including fluidic[12–14], (electro-)chemical[15,16], and thermal[17–20] actuation. Among these technologies, soft electrostatic actuators driven by high electric fields combine excellent speed and efficiency, with self-sensing capabilities, a high level of controllability, portability, and facile integration with electronic control systems[21–23].

[1]Robotic Materials Department, Max Planck Institute for Intelligent Systems, Stuttgart, Germany. [2]Institute for Adaptive Mechanical Systems, University of Stuttgart, Stuttgart, Germany. [3]Paul M. Rady Department of Mechanical Engineering, University of Colorado Boulder, Boulder, CO, USA. [4]Materials Science and Engineering Program, University of Colorado Boulder, Boulder, CO, USA. ✉e-mail: philipp.rothemund@iams.uni-stuttgart.de; ck@is.mpg.de

Dielectric elastomer actuators (DEAs) are based on solid dielectrics in the form of elastomeric membranes sandwiched between compliant electrodes[24], and they are the first technology that highlighted the outstanding attributes of soft electrostatic actuators[25–28]. However, DEAs typically require stacked designs to achieve large linear contraction upon activation, necessitating specialized fabrication procedures using stretchable materials, and often suffer from limited reliability[11,29].

To address these limitations, soft electrostatic actuators based on solid-liquid architectures were introduced; the use of a second, liquid phase of dielectric, offers a higher level of design flexibility, it enables the use of flexible but inextensible polymer films, and it introduces the ability to self-heal from dielectric breakdown events[30]. Especially, soft electrohydraulic actuators further utilize the liquid dielectric for coupling electrostatic and hydraulic forces; this concept of electrohydraulic amplification has been demonstrated in hydraulically amplified self-healing electrostatic (HASEL) actuators, which offer versatile actuation modes[30–33] while featuring all-around actuation performance rivaling biological muscles[11,30]. HASELs have been demonstrated in various applications, spanning from biomimetic underwater robots and musculoskeletal jumping robots to wearable haptic devices[34–39]. Other examples of electrohydraulic actuators include electrostatic bellow muscle (EBM) actuators, which demonstrated additional energy harvesting capabilities[22], and hydraulically amplified taxels (HAXELs), which enabled dense arrays of small actuators for wearable haptic devices[40–42]. Unlike electrohydraulic actuators, Rossiter et al. showed that limited use of liquid dielectric in the zipping region without utilizing hydraulic amplification can initiate and drive electrostatic actuation upon voltage application, resulting in very lightweight actuators with open structures[43–45]. However, this design cannot tap into the benefits of electrohydraulic amplification, and the open structures make it challenging to retain the liquid dielectric, potentially reducing the robustness of actuation. Electrohydraulic actuators, conversely, face challenges from the substantial mass of the liquid dielectric affecting crucial mass-based performance metrics, such as specific energy and specific power. One proposed approach to reduce mass while maintaining performance involves miniaturizing electrohydraulic actuators[37,46]; however, downscaling comes with its own challenges, including mechanical constraints from film stiffness, fringe-field effects, and fabrication complexities at smaller scales, ultimately limiting overall performance[46].

Here, we present ultralight soft electrostatic actuators based on solid-liquid-gas architectures, where we introduce a third phase into the solid-liquid structures of electrohydraulic actuators in the form of a gaseous dielectric that replaces a large portion of the liquid dielectric. Through a combination of theoretical analysis and experimental validation, we show that Paschen's law[47,48] can be used to determine the fundamental limits of actuator performance using this strategy, predicting a maximum ratio of gaseous dielectric for a given set of actuation conditions and materials properties of the dielectrics. We used the Peano-HASEL actuator[32], a type of soft electrohydraulic actuator that linearly contracts upon actuation, as a model system to study the fundamental mechanisms[46,49,50] and to demonstrate the benefits of using the solid-liquid-gas architecture. The resulting three-phase soft electrostatic actuators achieved outstanding performance metrics when ambient air was used as the gaseous dielectric—a specific energy of 33.5 J kg$^{-1}$ and a power-to-weight ratio of 1600 W kg$^{-1}$, which translates to a 532% and 1130% improvement, respectively, compared to the conventional Peano-HASEL actuators. Additionally, the three-phase actuators achieved an 83% higher strain rate (2840 % s$^{-1}$) and a 70% wider bandwidth (c.a. 100 Hz), indicating a substantial improvement of actuation speed. We showcased the outstanding performance of these actuators in a jumping robot, which exhibited a 60% increase in jump height and a 32% reduction in take-off time when compared to

its solid-liquid counterpart. We also identified a gas mixture of $C_4F_7N$ and $CO_2$ with particularly high dielectric strength, that enabled outstanding specific energy of 51.4 J kg$^{-1}$ (a nine-fold improvement over conventional Peano-HASELs), underlining a key advantage of implementing the proposed solid-liquid-gas architectures in closed structures that can enclose gaseous dielectrics with tailored properties.

## Results
### Model system based on solid-liquid-gas architecture
We used the Peano-HASEL actuator as a model system to explore the concept of solid-liquid-gas architecture (Fig. 1). The conventional Peano-HASEL actuator, based on a solid-liquid architecture (Fig. 1a, left side), consists of thin, flexible, but inextensible solid dielectric shells forming a pouch. This pouch is partially coated with flexible electrodes and filled with liquid dielectric. When a voltage $\Phi$ is applied across the electrodes, the attractive electrostatic force progressively closes the electrodes. This "zipping" motion displaces the liquid dielectric, which in turn generates a contractile strain capable of lifting loads[32]. Throughout this process, the liquid dielectric serves a dual purpose: sustaining high electric fields and hydraulically transmitting the electrostatic force (Pascal's principle), making it essential for this actuation mechanism. However, in the solid-liquid architecture, the liquid dielectric constitutes a large portion of the actuator mass $m_{act}$, with only thin and lightweight solid dielectric shells surrounding it; nearly 90% of the $m_{act}$ is liquid (proportional bars in the center of Fig. 1a) for the most widely used length scale, with pouch width and length in the cm-range. Exploring solid-liquid-gas architectures, by replacing an equivalent volume of liquid dielectric with a gaseous dielectric, offers a substantial reduction of the $m_{act}$ (Fig. 1a, right). However, for a gaseous dielectric to take over the dual role of liquid dielectrics in sustaining high electric fields and transmitting the electrostatic force, its lower breakdown strength[43,51] and compressibility may alter the actuation mechanism, potentially influencing the performance of gas-containing actuators.

To evaluate how different combinations of dielectric phases influence actuator response, we examined various configurations of liquid and gaseous dielectrics. We used a conventional solid-liquid Peano-HASEL actuator as a reference, then fabricated solid-liquid-gas and solid-gas variants for comparison (Fig. 1b and Supplementary Movie 1; see Methods section for details on fabrication and experimental setup). Air served as the gaseous dielectric due to its accessibility, non-flammability, and non-toxicity. Under a representative actuation condition (7 kV actuation voltage and 1 N applied load), the solid-liquid-gas actuator demonstrated a comparable actuation stroke to the reference solid-liquid actuator ($\chi_2 \approx \chi_1$) while featuring an 84% mass reduction ($m_1 = 1.9$ g, $m_2 = 0.3$ g), thereby highlighting an important benefit of using the solid-liquid-gas architecture. In contrast, the solid-gas actuator failed to actuate ($\chi_3 = 0$), despite its drastic decrease in actuator mass ($m_3 = 0.2$ g).

As shown in Fig. 1c, we attribute this lack of actuation in the solid-gas actuator to premature dielectric breakdown in the gas, due to the lower dielectric strength of gaseous dielectrics compared to liquid dielectrics. This figure qualitatively illustrates the strength of the electric field and the proximity to dielectric breakdown in both liquid and gaseous dielectrics. In all configurations, the electric field peaks near the "zipping front", where the zipping region propagates, and rapidly decays away from it. The liquid dielectric in the solid-liquid configuration provides sufficient insulation against dielectric breakdown, sustaining the high electric fields required for actuation. On the other hand, the gaseous dielectric in the solid-gas configuration is prone to breakdown at the same voltage, preventing actuation. However, having a small amount of liquid dielectric near the zipping front, where the electric field is concentrated, effectively shields the actuator from breakdown, allowing stable actuation in the solid-liquid-gas configuration, as shown in Fig. 1b.

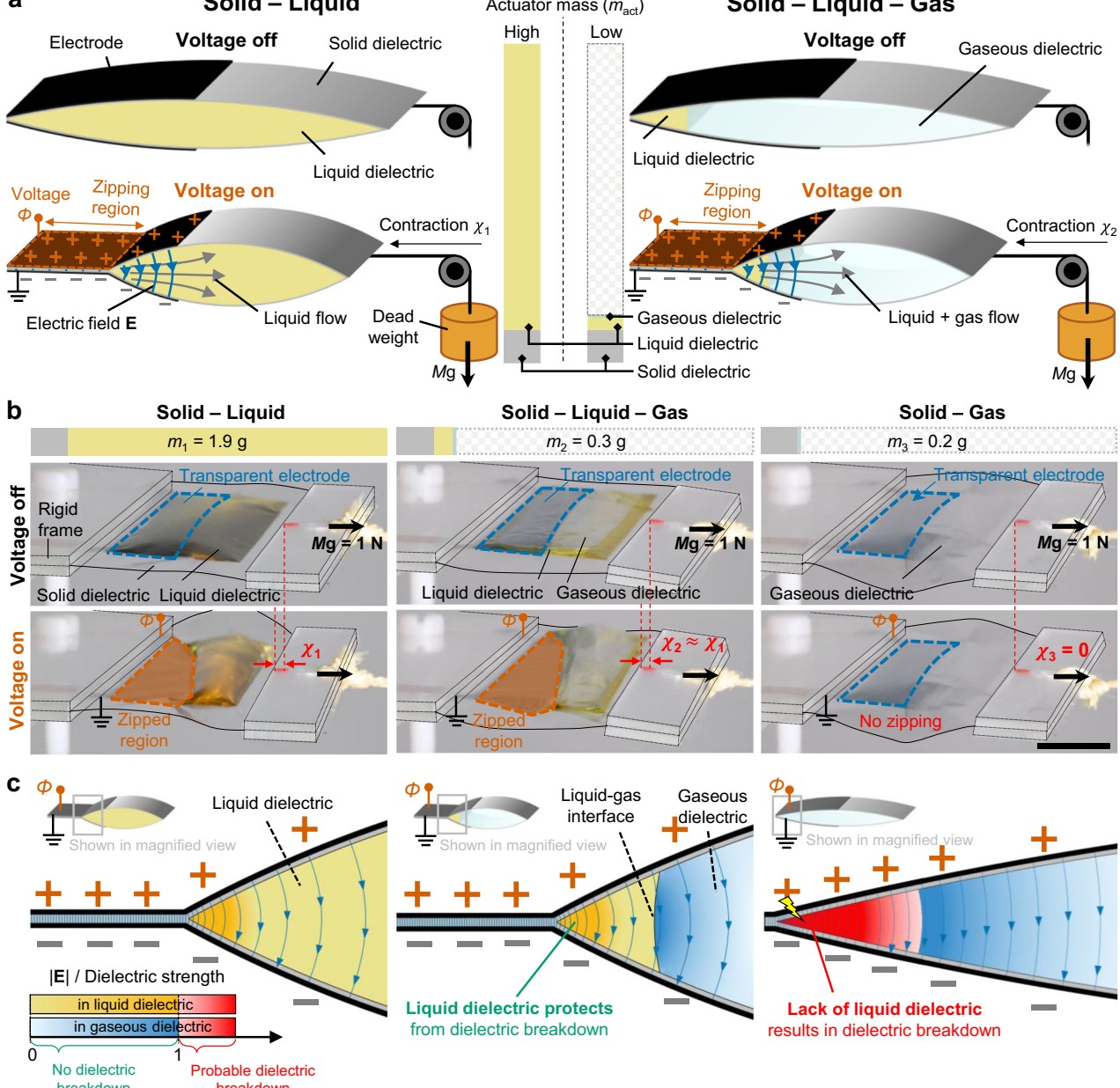

**Fig. 1 | Soft electrostatic actuators based on solid-liquid-gas architectures.** **a** Comparison of the actuation principles of Peano-HASEL actuators, used here as a model system, based on solid-liquid (left) and solid-liquid-gas (right) architectures, highlighting the potential for substantial reduction in actuator mass (center, proportional bars). When a voltage $\Phi$ is applied across the electrodes, the electrostatic force "zips" the solid dielectric shells together, displacing the liquid or gaseous dielectric within, thus generating a contractile strain. **b** Photographs of actuators based on different dielectric architectures; solid-liquid and solid-liquid-gas actuators demonstrated similar actuation stroke ($\chi_2 \approx \chi_1$), while the solid-liquid-gas actuator featured an 84% reduction in mass. The solid-gas actuator failed to actuate ($\chi_3 = 0$). Scale bar, 1 cm. **c** Schematic of electric field and probability of dielectric breakdown near the zipping front. In solid-liquid and solid-liquid-gas actuators, the liquid dielectric protects the region where the electric field is the highest from breakdown, while the solid-gas actuator is prone to breakdown due to the substantially lower dielectric strength of the gaseous dielectric.

In the following sections, we further examine the effects of gas inclusion, focusing on its lower dielectric strength and compressibility, to analyze the fundamental mechanisms that determine the maximum gas ratio allowed for preserving actuation performance under a given set of actuation conditions.

## Quasi-static performance

To determine the maximum gas ratio allowed, we examined quasi-static force-strain characteristics of actuators with varying volume fractions of air, ranging from 0% to 100%. As qualitatively illustrated in

Fig. 2a, sufficient liquid dielectric near the zipping front is expected to shield against dielectric breakdown, leaving compressibility of the gaseous dielectric as the other main factor potentially decreasing performance. In contrast, insufficient liquid dielectric should result in dielectric breakdown in air, preventing effective actuation.

To achieve quasi-static actuation, we applied a slow voltage ramp of $1 \, kV \, s^{-1}$, gradually reaching to a maximum voltage of 8 kV across the actuators at various constant forces. We then measured the resulting strains at the maximum voltage (Supplementary Fig. 1; see Methods section for details on fabrication and experimental setup). In the

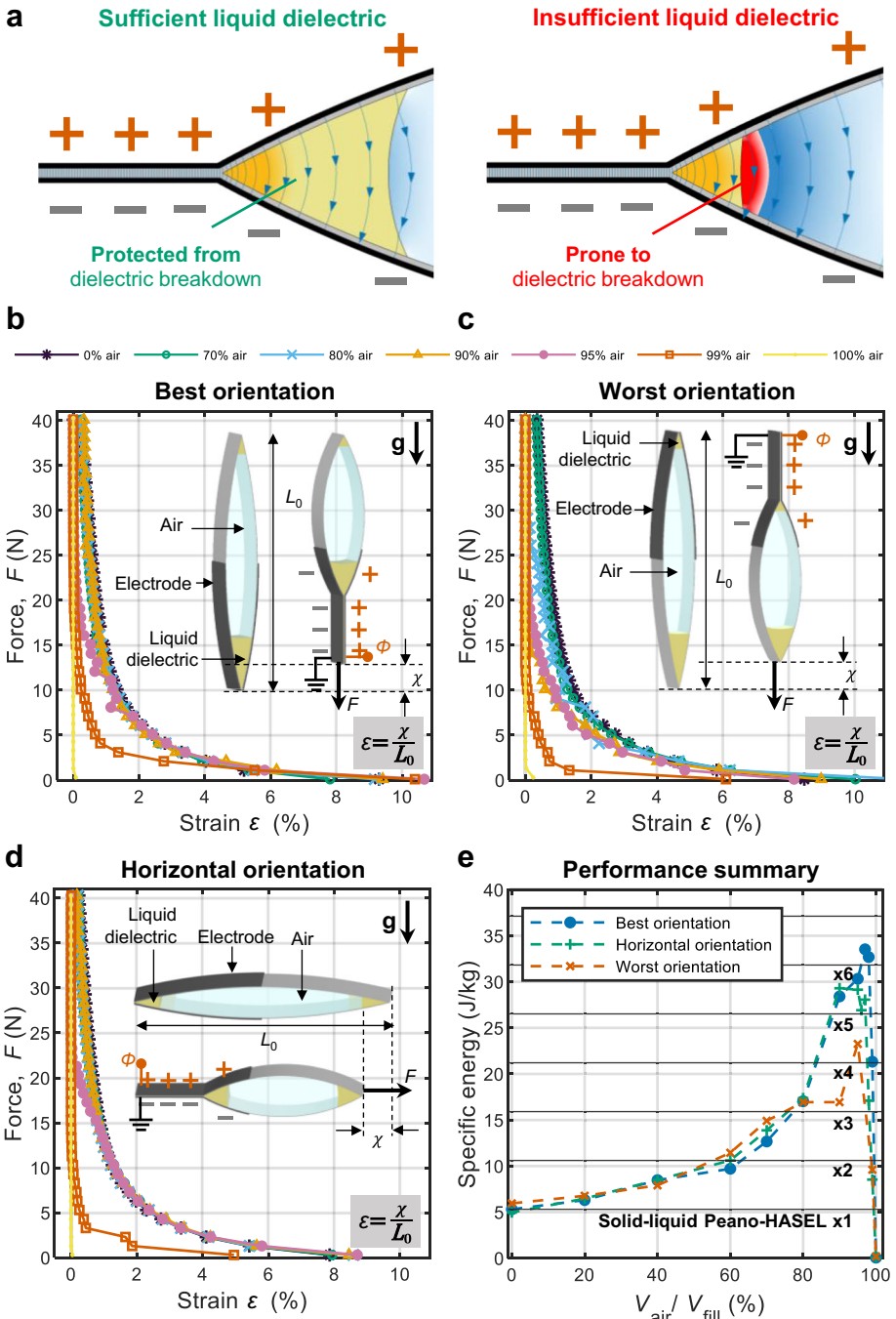

**Fig. 2 | Evaluation of quasi-static performance of Peano-HASEL actuators based on solid-liquid-gas architectures. a** Schematic of actuators with different amounts of liquid dielectric; sufficient liquid near the zipping front prevents dielectric breakdown, while insufficient liquid increases the probability of breakdown of the gaseous dielectric. **b** Force-strain curves of actuators with varying percentages of air-fill in the "best" orientation (electrodes at the bottom). Performance was stable up to 90% air-fill, then it declined abruptly, completely failing at 100% air-fill. **c** Force-strain curves of actuators with varying percentages of air-fill in the "worst" orientation (electrodes at the top). Performance was stable up to 70% air-fill (lower than the 90% limit observed in the best orientation), then declined abruptly, completely failing at 100% air-fill. **d** Force-strain curves of actuators with varying percentages of air-fill in the horizontal orientation. Performance was stable up to 90% air-fill, then it declined abruptly, completely failing at 100% air-fill. **e** Specific energy as a function of percentage of air-fill for the three orientations, with a peak of 33.5 J kg⁻¹ at 97% air-fill in the best orientation. Source data are provided as a Source Data file.

resulting force-strain plots (Fig. 2b), we observed no notable deviation in the curves spanning from 0% to 90% air-fill. However, as the percentage of air-fill increased beyond 90%, actuation strain rapidly decreased, with actuators eventually completely failing to actuate at 100% air-fill (see Supplementary Fig. 1 for an illustration of the behavior of an actuator with 95% air-fill). This result shows that, up to 90% air-fill, the inclusion of air has a negligible influence on force-strain

characteristics (and consequently work output), despite the lower breakdown strength and compressibility of air. To further investigate this experimental observation, we derived an analytical model of three-phase Peano-HASELs based on the solid-liquid-gas architecture, which accounts for the compressibility of gas (see Supplementary Fig. 2, and Supplementary Note), confirming that gas compressibility has negligible influence on the performance of the actuators investigated here

(see Supplementary Figs. 3, 4; we note that for future material systems that enable extremely high actuation forces, the compressibility of gas will start to affect actuation output). The model predicts negligible deviation between isentropic and isothermal extremes under typical actuation conditions (see Supplementary Fig. 3e); therefore, we assume isothermal conditions throughout this study without compromising the validity of the results. Combining these modeling results with our experimental observation, we hypothesize that the loss of actuation strain beyond 90% air-fill results from an insufficient amount of liquid dielectric near the zipping front.

To further investigate this hypothesis, we rotated the actuator by 180 degrees from an orientation where electrodes are located at the bottom ("best orientation", Fig. 2b) to another orientation where the electrodes are located at the top ("worst orientation", Fig. 2c). In this worst orientation, gravity pulls the liquid dielectric away from the zipping front, leaving only a small amount held in place by surface tension, for protection against dielectric breakdown. Still, the liquid dielectric consistently remains at the zipping front throughout actuation, regardless of the actuator orientation, as shown in Supplementary Fig. 5. The force-strain curves in Fig. 2c indicate that deviation from the fully liquid-filled reference curve starts to become noticeable above air-fill of 70% (lower than the 90% limit observed in the best orientation), supporting our hypothesis that insufficient liquid dielectric at the zipping front is the primary cause for performance loss at very high air-fill levels.

We tested the influence of air fill on force relaxation related to charge accumulation[52]. During prolonged actuation under constant voltage and force, the introduction of air had a negligible effect on charge accumulation behavior (Supplementary Note, Supplementary Fig. 6a).

To quantify performance in intermediate orientations, we tested actuators positioned horizontally (Fig. 2d). The force-strain responses closely matched those of the best orientation up to 90% air-fill. This result indicates that up to an air-fill of 90% the performance of actuators will be consistent for all orientations between the best and the horizontal orientation.

As a key parameter to assess the performance, we calculated specific energies (J kg⁻¹) of our actuators, defined as the area under force-strain curves (i.e., the maximum mechanical work in one actuation cycle) divided by the actuator mass. As shown in Fig. 2e, the specific energy increased with higher air-fill ratios, peaking at a certain point. Notably, in the best actuator orientation, specific energy reached a maximum of $33.5 \, \text{J kg}^{-1}$ at 97% air-fill, representing a 532% enhancement over a reference actuator with 0% air-fill. In the worst actuator orientation, the specific energy reached a maximum of $23.3 \, \text{J kg}^{-1}$ at 95% air-fill, representing a 291% enhancement over a reference actuator with 0% air-fill in the same orientation. However, further increases in air-fill ratio beyond this point caused the specific energy to drop sharply to zero. The next section is devoted to understanding the fundamental mechanisms behind this performance drop, focusing on dielectric breakdown in gas as a key performance limitation.

## Identification of fundamental performance limits

To quantitatively examine the correlation between actuation performance and dielectric breakdown in air, we first established a theoretical threshold, based on Paschen's law, for predicting dielectric breakdown of air in our actuator. Paschen's law describes the breakdown voltage of gases between parallel metal electrodes, where an electron avalanche initiates dielectric breakdown in the gas[47]. Although factors such as electrode shape and dielectric layers shielding the electrodes may alter the breakdown characteristics of gas[51,53], we adopted Paschen's law as the foundation for our prediction, speculating that this mechanism still governs air breakdown in our actuator. Indeed, our experiments confirmed that actuator failure is

correlated with dielectric breakdown of air as predicted by Paschen's laws (Fig. 3).

To establish the theoretical threshold for predicting breakdown of air in our actuator, we integrated Paschen's law with the analytical model derived in the previous section (see Supplementary Figs. 2, 3, and Supplementary Note). This combined model is represented on a plane spanned by the x-axis of pressure times distance and the y-axis of voltage (hereafter we refer to this plane as the "Paschen's plane"), which provides an effective framework for analyzing the dielectric breakdown characteristics of gases (Fig. 3a). Paschen's curve for air, $\Phi_{\text{air}} = f(p, d)$[48], divides this plane into a "safe" region (no dielectric breakdown) and an "unsafe" region (dielectric breakdown in air may occur). Our model then maps the actuator status onto this plane, indicating whether the actuator is protected from dielectric breakdown or not. The representative model-derived line (blue line in Fig. 3a) shows the continuous change of the actuator status under a typical actuation scenario, as described in Fig. 3b. This scenario involves increasing the voltage $\Phi$ for an actuator with a given liquid volume $V_{\text{liq}}$ under a constant load $F$. Specifically, when given the set of values $(F, \Phi, V_{\text{liq}})$ describing actuation conditions, our model identifies the resulting actuator status $(p, d, \Phi_{\text{air}})$ and maps it onto the Paschen's plane. Here, $p$ is the air pressure inside the actuator, $d$ is the distance between solid dielectric shells along the liquid-gas interface (where breakdown is most probable), and $\Phi_{\text{air}}$ is the potential difference across the air gap at the liquid-gas interface (see Supplementary Fig. 7 for details). As the voltage $\Phi$ across the specific reference actuator shown in Fig. 3b is increased, the actuator encounters three distinct states, as marked with (a)-(c) in Fig. 3a: At point (a), the actuator does not exhibit zipping due to low voltage, so $p$ and $d$ remain at their initial values. Point (b) represents the state after the onset of zipping, where the actuator status lies in the safe region below Paschen's curve, allowing actuation. At point (c), $\Phi_{\text{air}}$ exceeds the critical threshold, causing dielectric breakdown of air and thus actuation failure. This critical threshold $\Phi_B$, which predicts impending breakdown of air, can be identified by locating the intersection between Paschen's curve, $\Phi_{\text{air}} = f(p, d)$, and the model-derived blue line ($p \cdot d$, $\Phi_{\text{air}}$), as determined by the following equations (see Supplementary Figs. 2, 7, and Supplementary Note for derivation):

$$p\left(F, \Phi, V_{\text{liq}}\right) = p_{\text{ext}} + \frac{F}{w(L_p - l_e\left(F, \Phi, V_{\text{liq}}\right))} \left(\frac{\alpha(F, \Phi)}{\cos[\alpha(F, \Phi)]}\right) \quad (1)$$

$$d\left(F, \Phi, V_{\text{liq}}\right) = (L_p - l_e\left(F, \Phi, V_{\text{liq}}\right)) \\ \frac{(\cos[\theta(\alpha(F, \Phi), l_e\left(F, \Phi, V_{\text{liq}}\right), V_{\text{liq}})] - \cos[\alpha(F, \Phi)])}{\alpha(F, \Phi)} \quad (2)$$

$$\Phi_{\text{air}}\left(F, \Phi, V_{\text{liq}}\right) = \frac{d\left(F, \Phi, V_{\text{liq}}\right)\varepsilon_s}{2t\varepsilon_{\text{air}} + d\left(F, \Phi, V_{\text{liq}}\right)\varepsilon_s}\Phi \quad (3)$$

For the above formulas we assume 1) a flat meniscus and 2) a uniform electric field, treating the solid dielectric shells as parallel plates at a distance $d$, with the effective potential across air $\Phi_{\text{air}}$ at the liquid-gas interface. Here, $w$ and $L_p$ are the width and length of a single actuator pouch, $l_e$ is the zipped length of electrodes, $\alpha$ is half of the central angle of the arcs in the simplified cross section of the actuator, $\theta$ describes the location of the liquid-gas interface, $\varepsilon_s$ and $\varepsilon_{\text{air}}$ are the dielectric constants of the solid dielectric and air, respectively, and $t$ is the thickness of the solid dielectric shell (see Supplementary Fig. 7 for parameter details).

Now we map the predicted breakdown points, where Paschen's curve intersects with the model-derived line (blue line in Fig. 3a) for

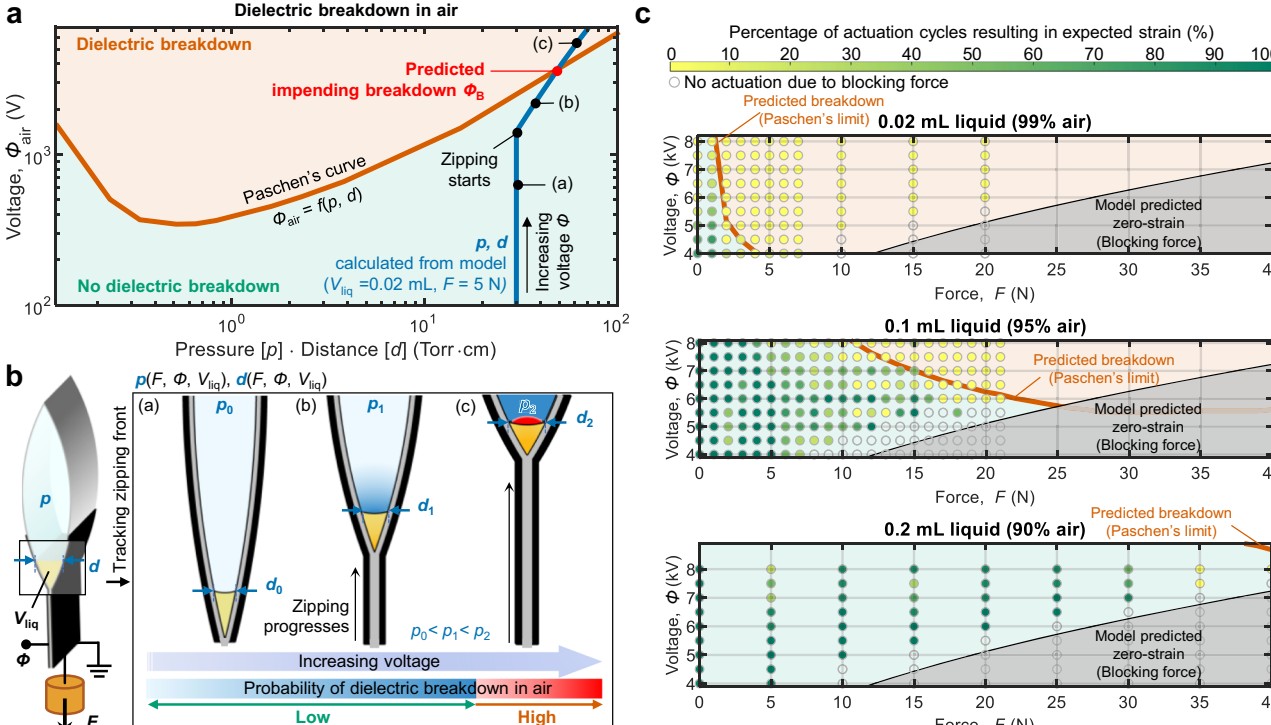

**Fig. 3 | Fundamental performance limits governed by Paschen's law. a** Paschen's curve for air (orange) and a model-derived line (blue), describing calculated actuator status as a function of voltage $\Phi$, shown on the "Paschen's plane"; the intersection of the orange and blue lines indicates the predicted impending dielectric breakdown of air within the actuator. **b** Schematics of a three-phase Peano-HASEL actuator at increasing voltage $\Phi$, showing three key states corresponding to the points marked in Fig. 3a: (**a**) before zipping starts; (**b**) zipped but below Paschen's curve, thus protected from breakdown; and (**c**) zipped surpassing Paschen's curve, thus prone to breakdown. **c** Theoretical predictions of breakdown (Paschen's limit is shown as an orange curve; the orange region above the curve indicates predicted breakdown, and the green region below the curve indicates states protected from breakdown) mapped onto the force-voltage plane. Experimental results are visualized using a color gradient spanning from green to yellow to indicate the gradual transition from successful actuation to failed actuation; the experimentally observed region of high probability of successful actuation (predominantly green-filled circles) is separated by the theoretically predicted boundary from a region of low probability of successful actuation (predominantly yellow-filled circles). Source data are provided as a Source Data file.

various conditions of $F$ and $V_{liq}$, onto the $F$-$\Phi$ plane, which describes actuation conditions. For a given $V_{liq}$, these breakdown points form a line on the $F$-$\Phi$ plane, represented as an orange line in Fig. 3c (Paschen's limit). This line now divides the $F$-$\Phi$ plane into a safe region (no dielectric breakdown) and an unsafe region (dielectric breakdown in air). The gray region indicates model predicted zero-strain, for conditions where the actuator has reached its blocking force[46]. We validated our model with experiments using quasi-statically actuated actuators with air-fill ratios of 90%, 95%, and 99% (see Methods section for details on fabrication and experimental setup). At constant forces, we slowly ramped the voltage and measured strains, recording failures when the measured strain fell below expected values, relative to reference actuators without air-fill (see Supplementary Fig. 8 for details on failure criteria). In Fig. 3c we mark data points where no actuation was observed due to blocking force by gray circles, data points with successful actuation by green-filled gray circles, and data points with failed actuation by yellow-filled gray circles. As dielectric breakdown is subject to statistical variations, we visualize the transition from successful actuation to failed actuation using a color gradient spanning from green to yellow, see Fig. 3c. Overall, our experimental results generally align well with the theoretical predictions: the experimentally observed region of high probability of successful actuation (predominantly green-filled circles) is separated by the theoretically predicted boundary, from a region of low probability of successful actuation (predominantly yellow-filled circles). The good agreement between experimental results and our modeling approach, combining Paschen's law with our analytical actuator model, confirms our hypothesis that the onset of electrical breakdown in the gas phase

governs the upper limit for air-fill ratio. This hypothesis was further confirmed with a simplified experimental setup with measured $p$ and $d$ in an open system (Supplementary Fig. 9; see Methods for details).

## Dynamic performance

In the previous sections, we examined the effects of air inclusion on quasi-static actuation. In this section, we present experimental results to investigate the effects of air inclusion on high-speed and high-frequency actuation. Specifically, we examined the step response and frequency response of three-phase Peano-HASEL actuators with air-fill ratios varying from 0% to 90%. All tests were conducted in the best orientation (electrodes located at the bottom of the actuator).

For the step response analysis, we applied a step voltage input with $\Phi_{peak} = 8$ kV to actuators loaded with deadweights ranging from 0.1 N (10 g) to 4.9 N (500 g). Displacement data was recorded using a high-speed camera (Fig. 4a; see Methods section for details on fabrication and experimental setup). The step response featured an accelerating phase, marking the initial period of actuator contraction, until the point where acceleration reaches zero ($\ddot{\chi} = 0$); this accelerating phase can be used to evaluate actuator performance (see Supplementary Fig. 10). Fig. 4b illustrates the resulting displacement data during the accelerating phase, at a representative actuation condition of $\Phi_{peak} = 8$ kV and deadweight of $Mg = 0.1$ N. Particularly, the speed rose abruptly from actuators with 60% to 80% air-fill. Accordingly, we categorized actuators with 0, 20, 40 and 60% air-fill as the "liquid-dominant group", and those with 80% and 90% air-fill as the "gas-dominant group". In supplementary information (see Supplementary Fig. 11 and Supplementary Note), we show that this speed increase

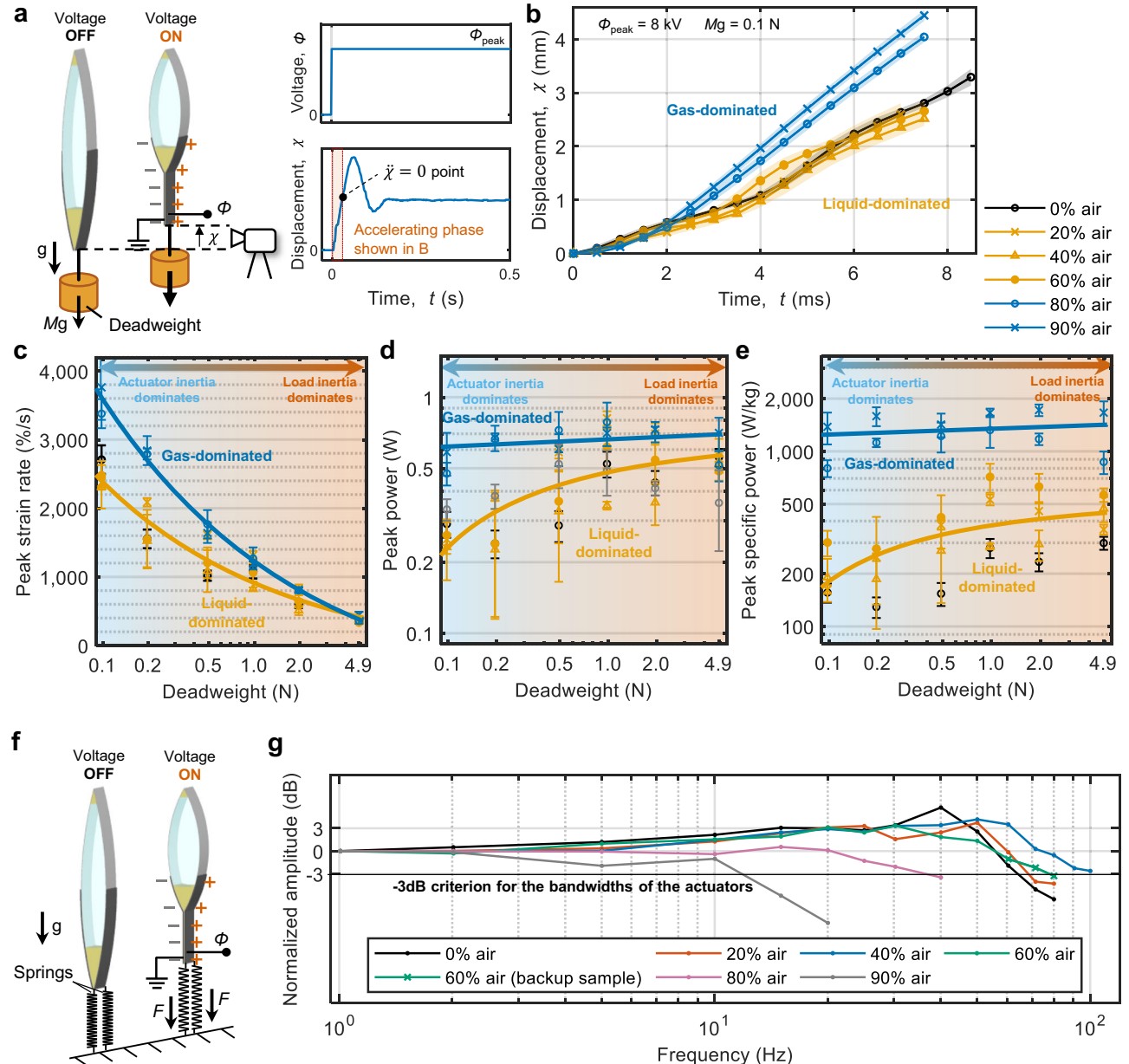

**Fig. 4 | Evaluation of dynamic performance of Peano-HASEL actuators based on solid-liquid-gas architectures. a** Experimental setup to evaluate actuator speed and power in response to a step input, with actuators loaded with deadweights. The shaded region in the time-displacement plot represents the initial acceleration phase. **b** Time-displacement plot of actuators with varying air-fill ratios, showing an abrupt increase in response speeds for actuators with air-fill ratios of 80% and above. Accordingly, actuators are categorized into "liquid-dominant" and "gas-dominant" groups. **c** Peak strain rate of actuators with varying air-fill ratios, with the gas-dominant group showing faster peak strain rates, especially with lighter deadweights, where the inertia of the actuator dominates the inertia of the external load. **d** Peak power of the actuators, with the gas-dominant group exhibiting higher power, particularly with lighter deadweights. **e** Peak specific power of the actuators, showing a substantial increase for the gas-dominant actuators due to their reduced weight and improved power. **f** Experimental arrangement used for the characterization of actuator bandwidth, with actuators loaded with springs. **g** Normalized amplitude of the actuators, with bandwidth increasing from 60 Hz to 100 Hz when going from 0% to 40% air fill; air fills above 40% resulted in decreased bandwidth. We note that the actuator with 60% air-fill broke during testing, and a backup sample was used afterwards. Shadings (**a**) and error bars (**c**–**e**) represent standard deviations for $n \geq 5$ trials. Source data are provided as a Source Data file.

stems from reduced inertia of liquid rather than changes in the effective viscosity of liquid-gas mixtures, as all the tested actuators operate in the inertial rather than the viscous regime[50].

For a quantitative assessment of the response speed, we characterized the peak strain rates by taking the derivative of the measured displacement data (Fig. 4c and see Supplementary Fig. 10 for the detailed definition of the peak strain rate). The liquid-dominant and gas-dominant groups showed a noticeable difference in peak strain rates, especially for lower deadweights, with the difference diminishing for heavier loads. Overall, this diminishment in difference indicates

that there are two regions: one where inertia of actuator mass dominates (towards lower deadweights) and another one where inertia of external load dominates (towards higher deadweights); from a practical perspective, it implies that the speed-increase for air-filled actuators is particularly pronounced when lifting low external loads. Peak power values (Fig. 4d; see Supplementary Fig. 10 for the definition) for the liquid-dominant group rose gradually with increasing deadweight up to the maximum weight evaluated here; peak powers in the gas-dominant group exceeded values of the liquid-dominant group, especially in the region with low external loads. Importantly,

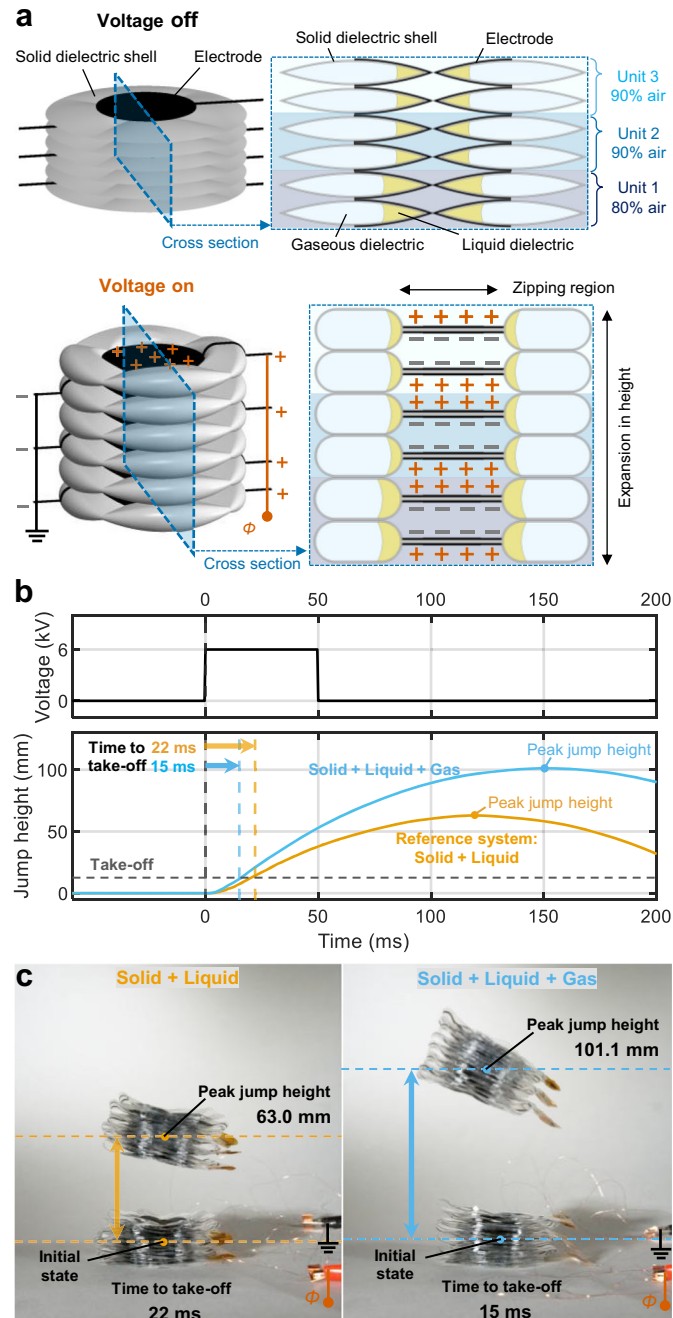

**Fig. 5 | Demonstration of agility of a jumping robot powered by soft electrostatic actuators based on solid-liquid-gas architectures. a** Illustration of a jumping robot and its cross-sectional view comprising multiple layers of quadrant donut-shaped HASEL actuators, arranged in 3 units, stacked vertically. This simplified schematic shows only two layers per unit, but the robot had four layers per unit. The ability of the robot to jump harnesses the explosive expansion in height of this structure in response to the application of a voltage step. **b** Comparison of jumping performance between robots based on solid-liquid and solid-liquid-gas architectures. A step voltage input lasting for 50 ms was applied to both robots. The solid-liquid-gas robot achieved a 60% higher jump height and a 32% shorter take-off time, demonstrating enhanced agility. **c** Photos of the robots at the initial state and at peak jump height. Scale bar, 2 cm. Source data are provided as a Source Data file.

the reduced actuator mass resulted in substantially higher peak specific powers in the gas-dominant group across the entire range of deadweights (Fig. 4e).

To investigate frequency response, we applied sinusoidal voltage inputs with $\Phi_{peak} = 6.0$ kV. Actuators were connected to two springs

($k = 0.004$ N/m), designed to exert an approximately constant force of 1 N at any displacement during testing (Fig. 4f; see Supplementary Fig. 12 for evaluation of the springs, see Methods section for details on fabrication and experimental setup). Fig. 4g shows the normalized amplitude (see Supplementary Fig. 13 for the detailed definition of amplitude) over the tested frequency range. Actuators with up to 40% air-fill showed an increase in bandwidth (amplitudes above the -3dB threshold), approximately from 60 Hz to 100 Hz, which we attribute to reduced inertia. However, at air-fill ratios above 40% the bandwidth started to decrease again; we speculate that dynamic splashing of liquid dielectric caused a decrease of the volume of liquid dielectric near the zipping front at high actuation frequencies, leading to dielectric breakdown of air and decreased actuator performance (see Supplementary Movie 2 for visualization of frequency response). The amplitude of the 0% air-filled reference actuator increases with frequency, peaking around 40 Hz. We found that this behavior is related to the internal motion of the liquid within the pouch—an intrinsic dynamic characteristic of electrohydraulic actuators; for frequencies around 30 Hz the motion of the liquid dielectric inside the pouch becomes synchronized with the driving frequency of the actuator, amplifying displacement; for details see Supplementary Fig. 14.

Notably, three-phase Peano-HASELs demonstrated substantially higher response speed and specific power, particularly in gas-dominant actuators at relatively low loads where actuator inertia dominates. Specifically, at an air-fill ratio of 90%, the peak strain rate reached 3760 % s$^{-1}$ with a deadweight of 10 g (0.1 N), 40% higher than that of the reference actuator with 0% air-fill; the peak strain rate reached 2840 % s$^{-1}$ with a deadweight of 20 g (0.2 N), 80% higher than that of the reference actuator. Additionally, the peak specific power reached 1600 W kg$^{-1}$ with a deadweight of 20 g (0.2 N), 11.3 times higher than that of the reference actuator with 0% air-fill.

All analyses above were conducted in the best orientation; similar evaluations in the worst orientation are provided in supplementary information (see Supplementary Fig. 15 and Supplementary Note).

We assessed the long-term durability of the actuators through 100,000-cycle tests. All tested actuators survived 100,000 cycles without failure; we only observed a decay of actuator strains starting at cycle numbers beyond 1,000, which we attribute to charge retention effects[52] stemming from the use of single polarity signals in these tests. Importantly, the actuator without any gas filling exhibited a very similar decay of actuation strain over cycles, indicating that the use of gas filling does not substantially change the durability of actuators up to 100,000 cycles (see Supplementary Fig. 6b and Supplementary Note).

## Demonstration of agility of a jumping robot

To highlight the capabilities of three-phase soft electrostatic actuators, we demonstrated their performance in a jumping robot. As shown in Fig. 5, this robot consisted of multiple layers of quadrant donut-shaped HASEL actuators[54], arranged in three units, stacked vertically. As shown in the cross-sectional view, in contrast to Peano-HASELs, which contract in length, donut HASELs expand in height, with both geometries relying on the same electrostatic zipping mechanism. Here, the rapid height expansion of donut HASELs enabled jumping when a step voltage was applied.

Two robots were constructed with identical geometries, differing only in their dielectric architectures: one based on a solid-liquid design and the other one on a solid-liquid-gas design. To enhance stability and balance during jumping, we slightly increased the amount of liquid dielectric in the bottom unit (Unit 1) in the solid-liquid-gas design. The resulting total masses of the robots were 24.2 g and 5.0 g, respectively. The experimental procedures involved a step input of 6 kV, with the voltage ON state sustained for 50 ms (see "Methods" section for details on experimental setup). The robot with the solid-liquid-gas architecture achieved a 60% higher jump height with a 32% shorter take-off

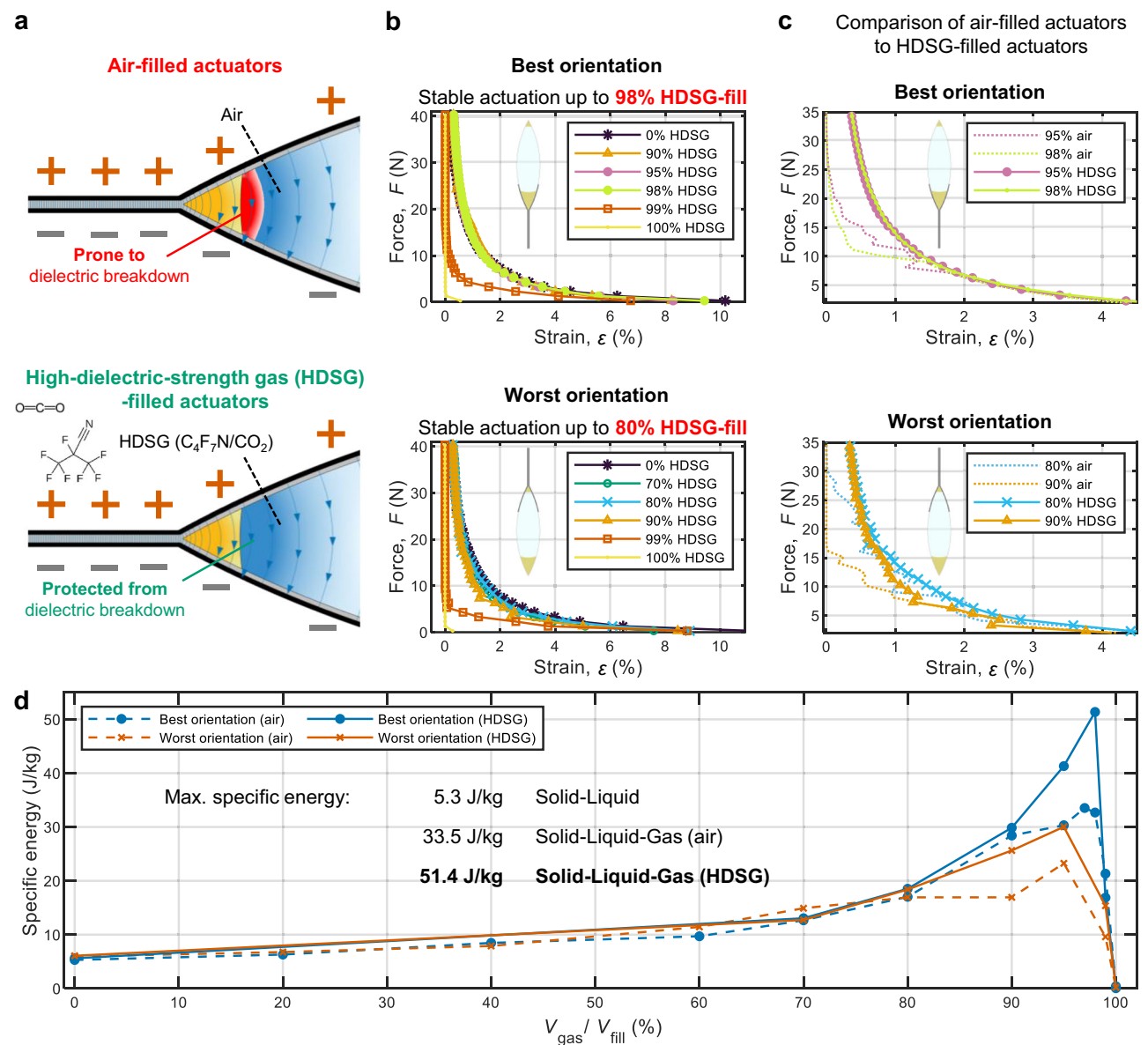

**Fig. 6 | Use of high-dielectric-strength gas (HDSG) enables actuators with improved quasistatic performance. a** Schematic of actuators with the same amount of liquid dielectric filled with air (left) and high-dielectric-strength gas (HDSG; right); a $C_4F_7N/CO_2$ gas mixture was used in this study. Dielectric strength of air is insufficient to prevent dielectric breakdown near the zipping front, while the HDSG provides sufficient insulation with the same amount of liquid dielectric. **b** Force-strain curves of actuators filled with HDSG in the best and worst orientation. The actuation was stable up to 98% HDSG-fill in the best orientation, improved from the 90% threshold with air. In the worst orientation, the actuation was stable up to 80% HDSG-fill, improved from the 70% threshold with air. **c** Comparison between air-filled and HDSG-filled actuators, in the best and worst orientations. **d** Corresponding specific energy of actuators in both best and worst orientations when filled with air or HDSG. In the best orientation, the maximum specific energy reached 51.4 J kg⁻¹ at 98% HDSG-fill, substantially higher than the specific energy of 33.5 J kg⁻¹ when using air. Source data are provided as a Source Data file.

time compared to the solid-liquid counterpart, highlighting the increased agility of the robot based on the three-phase design (Fig. 5b, c, and Supplementary Movie 3).

### High-dielectric-strength gas for enhanced performance in solid-liquid-gas architectures

As demonstrated above, a limitation of the performance of electrostatic actuators with solid-liquid-gas architectures is the dielectric breakdown of the gaseous phase. Closed electrostatic actuators, such as HASELs, HAXELs, and EBMs, offer the possibility of utilizing a high-dielectric-strength gas (HDSG) to improve performance considerably; HDSGs can prevent dielectric breakdown under conditions where air would typically fail (Fig. 6a). To demonstrate this idea, we used a HDSG in the form of a mixture of $C_4F_7N$ and $CO_2$ (see Methods for details),

which offers dielectric strength comparable to $SF_6$ but with substantially lower global warming potential[55,56]. To fill the actuators with HDSG, we used a glove box setup illustrated in Supplementary Fig. 16, to prevent contamination with ambient air (see Methods for details). With the method used for evaluating quasi-static performance as in Fig. 2 (see "Methods" for details), we showed that HDSG-filled actuators achieved stable operation up to 98% gas-fill in the best orientation and up to 80% in the worst orientation (Fig. 6b)—considerable improvements over the 90% and 70% thresholds observed with air, respectively. Fig. 6c provides a direct comparison of the force-strain curves of air-filled and HDSG-filled actuators. As a result, the maximum specific energy increased to 51.4 J kg⁻¹ at 98% HDSG-fill in the best orientation, and 30.0 J kg⁻¹ at 95% HDSG-fill in the worst orientation (Fig. 6d).

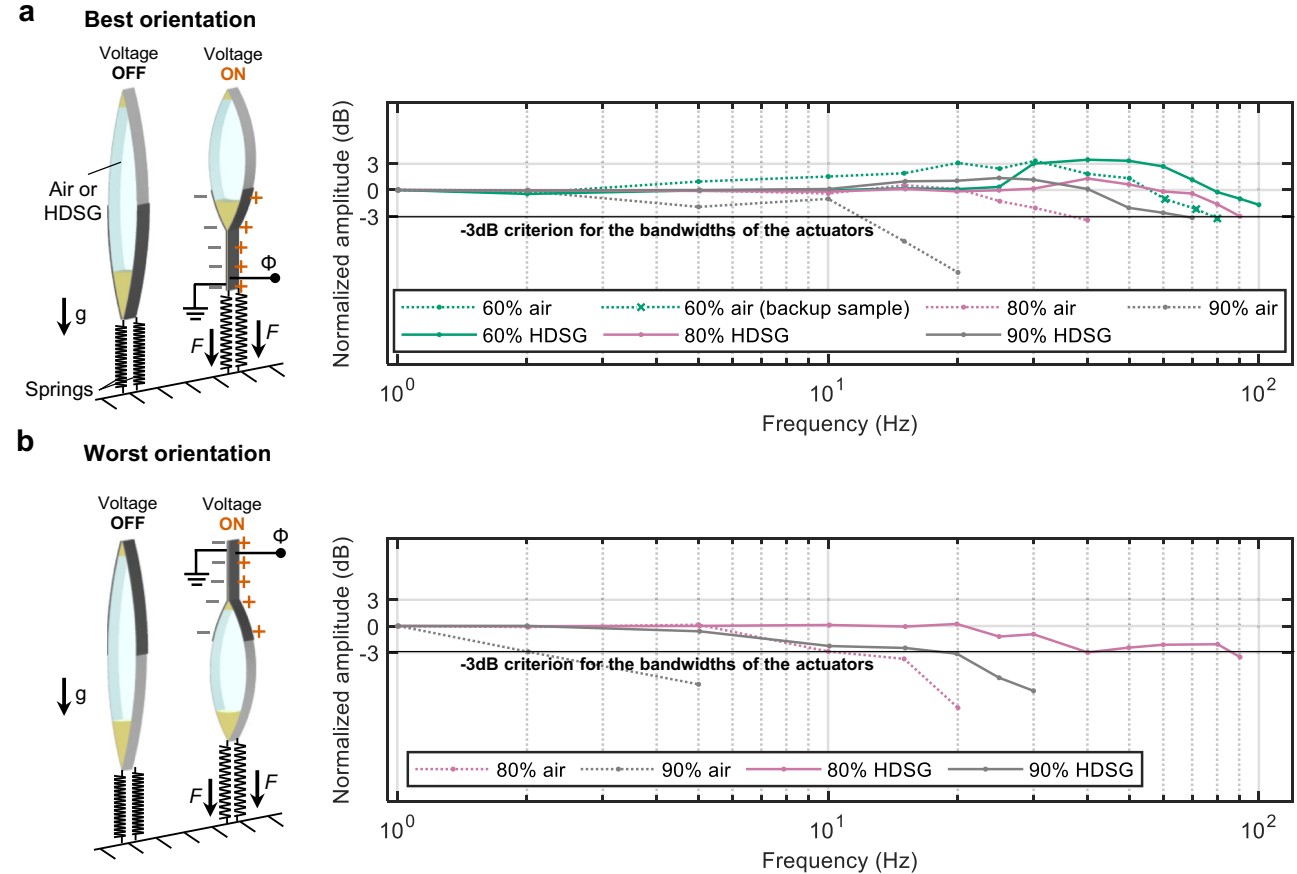

**Fig. 7 | Use of high-dielectric-strength gas (HDSG) enables actuators with improved bandwidth.** Normalized amplitude of the actuators filled with air (dashed lines) and high-dielectric-strength gas (HDSG; solid lines). Actuators filled with the HDSG ($C_4F_7N/CO_2$ gas mixture) exhibited increased bandwidth compared to the actuators filled with the same amount of air. **a** In the best orientation, actuators filled with 60% HDSG achieved a bandwidth above 100 Hz, improved

from 80 Hz of the 60% air-filled actuator. Similarly, 80% and 90% HDSG-filled actuators achieved improved bandwidths of 90 Hz and 70 Hz, compared to 40 Hz and 10 Hz for the corresponding air-filled actuators, respectively. **b** In the worst orientation, actuators filled with 80% and 90% HDSG achieved bandwidths of 80 Hz and 20 Hz, improved from 10 Hz and 2 Hz for the corresponding air-filled actuators, respectively. Source data are provided as a Source Data file.

We further evaluated the dynamic performance of HDSG-filled actuators using the same method used as in Fig. 4 (see "Methods" for details). As shown in Fig. 7, these actuators achieved broader bandwidths under cyclic voltage inputs in both the best and worst orientations compared with actuators having identical air-fill ratios. This improvement confirms that the use of HDSG successfully diminished the issue raised in the *Dynamic performance* section: at high actuation frequencies, dynamic splashing of the liquid dielectric reduces its local volume near the zipping front, leading to dielectric breakdown through air and lower performance. In contrast, actuator response to a step voltage input showed negligible improvement over the air-filled version (Supplementary Fig. 17).

## Discussion

We introduced ultralight soft electrostatic actuators based on a solid-liquid-gas architecture, demonstrating substantial performance improvements using Peano-HASELs as a model system. When filled with ambient air as the gaseous dielectric, the three-phase Peano-HASELs achieved a specific energy of 33.5 J kg$^{-1}$, narrowing the gap to the 40 J kg$^{-1}$ of biological muscles, and a specific power of 1600 W kg$^{-1}$, which far exceeds the 200 W kg$^{-1}$ typical of muscles[57,58]. This performance can be further enhanced by employing gases with tailored dielectric properties. Specifically, by using a gas mixture of $C_4F_7N$ and $CO_2$, which has a higher dielectric strength than air, the actuators reached a specific energy of 51.4 J kg$^{-1}$—surpassing that of biological muscle.

To highlight the potential of this architecture, we demonstrated a jumping robot using the solid-liquid-gas architecture. Compared to its solid-liquid counterpart, the robot achieved a 60% higher jump height with a 32% shorter take-off time, underscoring the ability of this architecture to enable powerful and agile soft robotic systems.

A key insight from our study is the identification of the fundamental performance limit of the solid-liquid-gas architecture, dictated by the onset of electrical breakdown in the gaseous phase. This limitation was validated through our analytical model for three-phase Peano-HASELs. This model, with the criteria set by Paschen's law, which governs dielectric breakdown in gases, exhibited excellent qualitative agreement with experimental results of actuation failures. The model predictions provided an upper bound on the ratio of the gaseous dielectric in three-phase Peano HASELs, for a given set of actuation conditions and materials properties of the dielectrics.

While Peano-HASELs, which have been successfully used in different robotics applications[37,59], served as a model system in this study, this work provides a foundational framework for the design and optimization of multi-phase electrostatic actuators in general. For instance, this approach can be easily expanded to other types of HASEL actuators[38,39,60], other types of soft electrohydraulic actuators, including electrostatic bellow muscle (EBM) actuators[22], hydraulically amplified taxels (HAXELs)[40–42], and other types of multi-phase soft electrostatic actuators, such as electro-ribbon actuators[43–45]. In closed actuators, the gaseous dielectrics can be actively engineered. Here, we have demonstrated that high-dielectric-strength gases can

substantially improve actuator stability and performance. Additionally, for applications where extremely lightweight actuators are critical, using larger-scale actuators that are filled with gases lighter than air to exploit buoyancy effects are an interesting direction of research —on the centimeter-scale, and below, however, the mass of the gas is negligible compared to the mass of the solid dielectric.

Our analytical model, which incorporates compressibility of gas, also provides insights into how actuator performance may vary under different environmental pressures and temperatures. Although the compressibility of the gas had negligible effects under standard conditions, performance may differ when the environmental conditions change (Supplementary Fig. 18a and b). Whereas the changes in actuation behavior are small for typical environmental variations, compressibility effects become more pronounced in extreme pressure and temperature environments. For example, high external pressure environments during underwater operation can noticeably compress the contained gas, severely altering the actuation behavior (Supplementary Fig. 18c). In such cases, however, the gas fill and the actuator design can be adjusted—pre-filling actuators at a higher pressure to match the external pressure, and extending electrodes length—to the targeted environmental operating conditions (Supplementary Fig. 18c). When the external pressure conditions undergo dynamic changes (such as when an underwater robot dives up and down), the force-strain characteristics of actuators would also undergo dynamic changes; in these cases, the control algorithms for the actuators would have to be adapted to account for these changes in actuation behavior.

Actuator orientation and dynamic effects were found to influence actuator performance by redistributing liquid dielectric within the pouch. This effect may compromise the practicality of these actuators in certain applications where stable liquid retention between the electrodes cannot be guaranteed. Several strategies could mitigate this effect. Selecting solid-liquid-gas combinations with higher surface tension (e.g., surface treatments or oleophilic metamaterials) can improve liquid retention near the pouch edges, counteracting gravitational or inertial forces. In addition, incorporating porous internal structures such as sponges or gels may help retain liquid in the desired places and prevent local depletion.

$C_4F_7N$ (also known as Novec™ 4710, 3M™) exhibits low acute inhalation toxicity (4-h $LC_{50}$ >10,000 ppm) and is not classified as carcinogenic, mutagenic, or reprotoxic[56,61]. Since the gas remains sealed within the actuator pouch, normal operation poses minimal risk when fabrication is performed in a well-ventilated area or fume hood. However, electrical breakdown can generate toxic decomposition byproducts (e.g., CO, $CO_2$, HF), so appropriate precautions are required during disposal or in the event of actuator failure.

In summary, the solid-liquid-gas architectures enable the development of ultralight, high-performing soft electrostatic actuators with excellent power-to-weight ratios, and these actuators can enable a range of robotic applications that require high agility, spanning from bioinspired robots for use in unstructured environments to soft wearable robotic systems and lightweight prostheses.

## Methods

### Details of actuator fabrication
Throughout the study, Peano-HASEL actuators were fabricated using the same solid-liquid-gas material system with identical geometric parameters. The process involved bonding solid dielectric—biaxially oriented polyester films (Mylar 850-15, Petroplast GmbH) using a CNC heat sealer[54] to create rectangular shells with a width $w = 6$ cm and a length $L_p = 2$ cm. Carbon ink, applied as flexible electrodes (Cl-2051, Nagase ChemteX America LLC), was screen printed on both sides of the shells. The shells were then filled with liquid dielectric (Silicone oil M5, Carl Roth) and gaseous dielectric (ambient air or HDSG), with a total volume of fluids ($V_{tot} = wL_p^2/4\pi$) chosen to theoretically form a cylinder when the electrodes are fully zipped[32]. Acrylic frames were

attached to the top and bottom of the shells for load application. These acrylic frames were not accounted for when measuring the actuator mass. For visualization purposes (in the section *Model system based on solid-liquid-gas architecture*; Fig. 1b and Supplementary Movie 1), a pigment (Beta-Carotene, SIGMA-ALDRICH Chemie GmbH) was added to the transparent silicone oil, and a transparent Pedot:PSS ink (Orgacon EL-P5015, Agfa-Gevaert N.V.) was used instead of the carbon ink.

We filled the actuators with the high-dielectric-strength gas ($C_4$-FN gas mixture: 20.00%-C4-FN; 2,3,3,3-tetrafluoro-2-(trifluoromethyl) propanonitrile and 80.00 %-$CO_2$; carbon dioxide; DILO Amaturen und Anlagen GmbH) in a glove box setup (GB0004; MSE Supplies®) to prevent contamination with ambient air (Supplementary Fig. 16).

### Experimental setup
In the experiments, the actuators were vertically suspended from the upper frame and the loads were attached to the lower frame. For visualization purposes, in the section *Model system based on solid-liquid-gas architecture*, the actuators were horizontally suspended and connected to a 100 g deadweight via a pulley system. A dual-mode lever system (310LR-C dual-mode lever arm system, Aurora Scientific Inc.) capable of simultaneously generating constant forces and measuring strains was used to apply loads for quasi-static analyses (Supplementary Fig. 1), deadweights for step response analysis and springs for frequency response analysis. Actuation strains were measured using the dual-mode lever system for the quasi-static analyses, a high-speed camera (Phantom v2640, Vision Research Inc.) tracking a marker (tracking software: Tracker) for the step response analysis, and a laser displacement sensor (LK-H157, Keyence) for the frequency response analysis. The input voltage signals were amplified by a high voltage amplifier (Trek® 610E, Advanced Energy Industries, Inc.). Another amplifier with a higher output current range (Trek® 50/12, Advanced Energy Industries, Inc.) was used for the dynamic analyses and the jumping robot demonstration.

### Fabrication and experimental details of the dielectric breakdown test in a simplified setup
Rigid acrylic plates were fixed at an angle of 27.6 ° relative to each other. Biaxially oriented polyester films (Mylar 850-15, Petroplast GmbH), each bearing 2 cm × 6 cm carbon electrodes (Cl-2051, Nagase ChemteX America LLC) printed on their outer surfaces, were securely bonded to the angled acrylic plates. The sides of the assembly were sealed using thin transparent acrylic plates. A varying amount of liquid dielectric (Silicone oil M5, Carl Roth) was added to fill the electrode gap. A voltage ramp was applied at a rate of $1\,kV\,s^{-1}$, with a 2-second hold at each 1 kV increment from 5 to 10 kV. This pattern was repeated for 100 cycles, alternating polarity each cycle. Using the measured current profiles, the breakdown voltages were recorded as the voltages at which the first current spikes occurred in each cycle.

### Generative AI
Generative AI software (ChatGPT, Open AI Inc.) was used in preparation of the manuscript, for improving flow of text.

## Data availability

Measured data and analysis code used to generate the main text figures are available in a public repository at https://doi.org/10.17617/3.83PLAQ. Source data are provided in this paper.

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

## Acknowledgements

This work was supported by the Max Planck Society, Germany. We thank the International Max Planck Research School for Intelligent Systems (IMPRS-IS) for supporting H.-J.J. and T.F.

## Author contributions

H.-J.J., P.R., and C.K. designed research; H.-J.J., T.F., and X.L. performed research; H.-J.J., T.F., X.L., A.S., S.J.A.K., and P.R. contributed new reagents/analytic tools; H.-J.J., T.F., and X.L. analyzed data; H.-J.J., T.F., X.L., A.S., S.J.A.K., P.R., and C.K. wrote the paper; P.R. and C.K. supervised the research.

## Funding

## Competing interests

C.K. is a coinventor on three patents, which cover the fundamentals and basic designs of HASEL actuators (assignee of all three patents is the Regents of the University of Colorado: US Patent 10995779B2, granted 2021-05-04; US Patent 11486421B2, granted 2022-11-01; and US Patent 11408452B2, granted 2022-08-09). C.K. is a cofounder of Artimus Robotics, a start-up company that commercializes HASEL actuators. The other authors declare that they have no competing interests.
