## [Transparent Peer Review file · Nature Communications]

Ultralight soft electrostatic actuators based on solid-liquid-gas architectures

Corresponding Author: Professor Christoph Keplinger

Version 0:

Reviewer comments:

Reviewer #1

(Remarks to the Author)

This study is the result of a series of ongoing studies of HASEL actuators by the authors group. I salute the authors for their ongoing research attitude.

In this study, the authors presented ultralight HASEL based on solid-liquid-gas architectures. As a results, the authors improved power-to-weight ratio by reducing actuator mass and increasing actuation speed. The key point is that the weight is reduced by replacing most of the dielectric liquid with air in HASEL. From the point of view of the mechanical performanice of the HASEL, the actuation performace was improved. However, the design of the HASEL is quite similar to previous Rossiter's works[43, 44, 45] as cited in the paper. Thus, there is little novelty in the design for new actuators. A few comments are noted below.

1. The purpose of this study is to improve the power-to-weight ratio by reducing actuator mass. Why didn't the authors investigate gases lighter than air?
2. The electrical breakdown is critical for electrostatic actuation. Why didn't the authors investigate gases with higher electrical breakdown voltages?
3. Have the authors verified experimentally that the simplified mathematical model is valid? What about static models in particular? What are the effects of gravity and how much deviation is there between the experiment and the ideal model in Supplementary Figure 2 ?
- 4.

Reviewer #2

(Remarks to the Author)

This work presents a novel soft electrostatic actuator based on a solid-liquid-gas structure, which demonstrates significant enhancement on the actuation performance of specific energy and power-to-weight ratio. It is interesting and brings some new perspectives to the electrostatic actuators. Some issues need to be addressed before publication.

1. It is assumed in the manuscript that liquid dielectric always exists at the zipping front due to surface tension. Can the authors use microscope photos or other pictures to prove this?
2. Will the maximum ratio of gaseous dielectric be influenced by the actuator orientation? If so, the limits should be discussed under the condition of worst orientation.
3. The effect of actuator orientations on actuation force and strain needs to be further discussed.
4. Does the composition or dielectric properties of the air have any impact on actuation behavior?
5. Long-term stability and reliability are critical for flexible actuators. Have the authors carried out cycle tests?
6. Since the liquid can move and oscillate in the air, the three-phase actuators are seem to be more unstable compared to the conventional Peano-HASELs. How to ensure the stability of the actuator output, especially in periodic high-frequency response?

Reviewer #3

(Remarks to the Author)

The manuscript presents an interesting approach to modifying the well-known Peano-HASEL actuator by partially replacing the liquid dielectric with a gas, forming a solid-liquid-gas architecture. The authors provide high quality and extensive experimental validation, including quasi-static tests, dynamic response analyses, and a jumping robot demonstration. The

proposed idea reduces mass and provide important improvements in performance metrics (specific energy, strain rate, power-to-weight ratio) of the Peano actuator; however, the conceptual novelty, solely connected to the partial filling approach, appears limited since the fundamental actuator design is already established in previous work (adequately cited in the manuscript).

An accompanying theoretical framework based on Paschen's law is introduced (which could potentially represent a valuable contribution) to explain the onset of dielectric breakdown, but there are some critical points regarding this.

Major Points:

1. Theoretical Model:

The current theoretical framework is quite simplistic; it uses Paschen's law with simplified assumptions (e.g., fixed geometry, flat meniscus, uniform field) to predict breakdown. However, the actual actuator features a complex interplay between liquid and gas, with dynamic variations such as variable gas gaps and trapped gas bubbles.

It is not clear how the static model maps onto the real dynamic fluid conditions observed (e.g., as seen in the supplementary videos). In this regard, the provided validation study, developed by testing the actuator as a whole, may not be effective. For a convincing validation, I suggest performing experiments on a dedicated, simplified, controlled setup aimed at isolating and validating the model parameters. This would help to limit the number of uncontrolled variables.

2. Thermal and Pressure Effects:

Although introducing air reduces mass, the inherent compressibility of gas could lead to unwanted effects when the actuator is subjected to temperature and pressure variations. The authors should address whether changes in these conditions might alter performance (e.g., stroke). A simple dedicated theoretical/experimental study examining the influence of thermal/pressure expansion on actuation behavior should be added. A discussion should also be introduced, highlighting possible connected limitations (e.g., underwater robotics?).

3. Long-term Durability:

The cyclical operational stability of the actuator should be shown, as differences may appear between the first and subsequent cycles due to changes induced in the interfaces between the fluid and gas. Thus, it would be valuable to see cyclical data and (possibly) preliminary discussion on degradation mechanisms that might occur over many cycles.

Minor Points:

- Assumption of Isothermal Conditions:

The model presented in the supplementary material applies Boyle's law ($pV = \text{constant}$) under the assumption of isothermal conditions. While this assumption may hold in quasi-static regimes, it should be clarified how dynamic conditions—where heat exchange may occur at a slower rate—could affect the model's validity.

- Force Relaxation Phenomena due to Charging:

It would be beneficial to know if any force relaxation related to charge accumulation behavior was observed during testing and whether the introduction of gas has any influence on such phenomena.

Version 1:

Reviewer comments:

Reviewer #1

(Remarks to the Author)

The paper maintains its novelty by adding the gas with high dielectric strength in response to the previous comment. However, Figures 3, 4, and 5 remain unchanged. Considering the overall scenario, experiments and considerations regarding gases with high dielectric strength should be added to ensure consistency.

Reviewer #2

(Remarks to the Author)

The authors have addressed my previous concerns. The paper is recommended for publication.

Reviewer #3

(Remarks to the Author)

The authors have responded to most of the questions and provided a substantial amount of additional work and experiments. In particular, the in-depth analysis of breakdown phenomena is a very valuable addition to the manuscript, enhancing both its novelty and impact. Moreover, the theoretical elaborations adequately address my previous comments.

There are, however, a few aspects that could be considered in a further minor revision:

1) Durability tests – Long-term durability tests have been conducted, achieving 100k cycles before degradation. The authors should provide a critical discussion of this result, as it appears to represent a worsening compared to the durability previously obtained with other multilayer liquid/solid actuators proposed by their group and by other research teams.

2) Bandwidth testing – I am moderately skeptical about the bandwidth tests. It is difficult to explain why an increase in amplitude is observed in most cases. For example, actuators with 0% air show an amplification of almost 6 dB at 40 Hz. Could this not be attributable to a structural resonance due to the mass of the attached frame? Wouldn't this compromise the evaluation of the actual bandwidth of the actuation system?

3) Use of C₄F₇N gas – The additional tests employing C₄F₇N gas are quite interesting. I would encourage the authors to comment on the possible toxicity of this gas, and whether they see any limitations in its use for their actuators, or conversely, if they consider it to be perfectly safe under all conditions without special precautions.

Version 2:

Reviewer comments:

Reviewer #1

(Remarks to the Author)

The paper is well revised and should be published.

Reviewer #3

(Remarks to the Author)

I confirm the authors have addressed my initial concerns and the manuscript is eligible for publication.

Highlights of major revisions

We would like to take this opportunity to thank all reviewers, who have given very insightful and constructive feedback, which we have addressed with a variety of new experimental results and major revisions; overall, we are excited to report that these revisions have substantially improved both the novelty and the quality of our manuscript.

Key updates include:

- Identification of C_4F_7N/CO_2 gas mixtures for use as high-dielectric-strength gas (HDSG); development of a fabrication procedure for HDSG-filled actuators that achieve substantially improved actuator performance (33.5 J kg^{-1} for air-filled actuators $\rightarrow 51.4 \text{ J kg}^{-1}$ for HDSG-filled actuators).
- Development of a new experimental setup with a simplified geometry to confirm Paschen's law of dielectric breakdown as the fundamental limit for gas-filling.
- Evaluation of actuator performance in an additional (horizontal) orientation.
- Evaluation of long-term stability and reliability; experimental demonstration of 100,000 actuation cycles without failure.
- Extension of the analytical model to include both isothermal and isentropic conditions, as well as evaluation of the effects of external pressure and temperature.

We have revised the manuscript to address all reviewer comments in detail. Below, please find a point-by-point response to all comments. We also enclose the revised manuscript, in which all changes are highlighted in yellow.

Reviewer 1

Reviewer 1, general comment:

“This study is the result of a series of ongoing studies of HASEL actuators by the authors group. I salute the authors for their ongoing research attitude.

In this study, the authors presented ultralight HASEL based on solid-liquid-gas architectures. As a result, the authors improved power-to-weight ratio by reducing actuator mass and increasing actuation speed. The key point is that the weight is reduced by replacing most of the dielectric liquid with air in HASEL. From the point of view of the mechanical performance of the HASEL, the actuation performance was improved. However, the design of the HASEL is quite similar to previous Rossiter's works[43, 44, 45] as cited in the paper. Thus, there is little novelty in the design for new actuators.”

Response:

As the reviewer points out, the performance improvement enabled by the solid-liquid-gas architecture is one important outcome of this paper. Additionally, the paper features two novel aspects, which we feel are important advances over existing literature; we have revised the manuscript to more clearly articulate these advances:

- 1) HASEL actuators and the actuators developed by Rossiter and colleagues share liquid-enhanced electrostatic zipping as an actuation mechanism. However, Rossiter's actuators are open structures, which directly interact with the surrounding air, whereas HASELs based on the solid-liquid-gas architecture are closed structures, where air can be replaced with other gases that feature high dielectric strength. Stimulated by insightful comments of reviewers 1 and 2, in the process of this major revision, we have identified a C_4F_7N/CO_2 gas mixture for use as high-dielectric-strength gas (HDSG); we have further developed a fabrication procedure for HDSG-filled actuators that achieve substantially improved actuator performance (33.5 J kg^{-1} for air-filled actuators $\rightarrow 51.4 \text{ J kg}^{-1}$ for HDSG-filled actuators; see detailed discussion in **Reviewer 1, comment 2**).
- 2) Building upon Paschen's law, we derived a model to predict dielectric breakdown through gas in HASELs based on the solid-liquid-gas architecture; through theoretical and experimental analyses, we pinpoint the fundamental performance limit as the electrical breakdown in the gas, thereby providing a guideline for selection of gaseous dielectrics. The model introduced in the manuscript should, in principle, also be applicable for determining the minimum amount of liquid in Rossiter's actuators. To confirm that the model also works for open structures, we performed

dielectric breakdown tests of liquid-gas interfaces in a simplified setup (see detailed discussion in **Reviewer 3, comment 1**)

Reviewer 1, comment 1:

“The purpose of this study is to improve the power-to-weight ratio by reducing actuator mass. Why didn't the authors investigate gases lighter than air?”

Response:

The use of gases lighter than air indeed reduces the weight of the actuator. However, our calculations indicate that the mass of the gas is negligible compared to the liquid and solid dielectrics, when considering the centimeter-scale we are currently operating at (even at an air-fill ratio of 100%, the mass of the gas accounts for less than 2% of the total weight). Still, for larger-scale actuators with substantial gas volumes, buoyancy effects could become more relevant. Using gases lighter than air, thus could be an interesting approach for applications where extremely lightweight actuators are critical.

To reflect this point, we have revised the **Discussion** section as follows:

Main text, page 15, line 351 (**Discussion**): “Additionally, for applications where extremely lightweight actuators are critical, using larger-scale actuators that are filled with gases lighter than air to exploit buoyancy effects are an interesting direction of research—on the centimeter-scale and below, however, the mass of the gas is negligible compared to the mass of the solid dielectric.”

Reviewer 1, comment 2:

“The electrical breakdown is critical for electrostatic actuation. Why didn't the authors investigate gases with higher electrical breakdown voltages?”

Response:

We appreciate this very insightful suggestion. As described above in our response to Reviewer 1 general comment, a key advantage of HASELs based on the proposed solid-liquid-gas architecture—enabled by its closed structure—is the possibility of using gaseous dielectrics with tailored properties to enhance electrostatic actuation.

Our model suggests that using high-dielectric-strength gas (HDSG) can lead to considerable performance gains, and thus motivated us to invest substantial experimental efforts to develop a fabrication procedure for HDSG-filled actuators that promise to achieve substantially improved actuator performance. We are excited to report that these efforts have resulted in a substantial improvement of actuator performance: from 33.5 J kg⁻¹ for air-filled actuators to 51.4 J kg⁻¹ for HDSG-filled ones, even surpassing the

performance of mammalian muscle (c.a. 40 J kg^{-1}) [R1]. For these experiments, we used a $\text{C}_4\text{F}_7\text{N}/\text{CO}_2$ gas mixture as the HDSG, which offers dielectric strength comparable to SF_6 but with substantially lower global warming potential [R2, R3]. During quasi-static actuation, the HDSG-filled actuators enabled reliable actuation up to 98% fill in the best orientation and up to 80% fill in the worst orientation—considerable improvements over the 90% and 70% thresholds observed with air, respectively.

We added these results as a new Figure 6, which shows a direct performance comparison between air-filled and HDSG-filled actuators, to the manuscript. We also added a new section *High-dielectric-strength gas for enhanced performance in solid-liquid-gas architectures* and revised **Abstract**, **Introduction**, **Discussion**, and **Methods** to implement the new points and findings as shown below.

[R1] J. D. Madden et al., *Artificial muscle technology: physical principles and naval prospects*, IEEE J. Ocean. Eng. 29, 706-728, 2004

[R2] M. Rabie et al., *Assessment of eco-friendly gases for electrical insulation to replace the most potent industrial greenhouse gas SF_6* , Environmental Science & Technology, 2018.

[R3] J. Owens et al. *Recent development of two alternative gases to SF_6 for high voltage electrical power applications*, Energies, 2021.

Main text, page 13, line 310:

“*High-dielectric-strength gas for enhanced performance in solid-liquid-gas architectures*

As demonstrated above, a limitation of the performance of electrostatic actuators with solid-liquid-gas architectures, is the dielectric breakdown of the gaseous phase. Closed electrostatic actuators, such as HASELS, HAXELs, and EBMs, offer the possibility of utilizing a high-dielectric-strength gas (HDSG) to improve performance considerably; HDSGs can prevent dielectric breakdown under conditions where air would typically fail (Fig. 6a). To demonstrate this idea, we used a HDSG in the form of a mixture of $\text{C}_4\text{F}_7\text{N}$ and CO_2 (see Methods for details), which offers dielectric strength comparable to SF_6 but with substantially lower global warming potential^{55,56}. With the method used for evaluating quasi-static performance as in Fig. 2 (see Methods for details), we showed that HDSG-filled actuators achieved stable operation up to 98% gas-fill in the best orientation and up to 80% in the worst orientation (Fig. 6b)—considerable improvements over the 90% and 70% thresholds observed with air, respectively. Fig. 6c provides a direct comparison of the force-strain curves of air-filled and HDSG-filled actuators. As a result, the maximum specific energy increased to 51.4 J kg^{-1} at 98% HDSG-fill in the best orientation, and 30.0 J kg^{-1} at 95% HDSG-fill in the worst orientation (Fig. 6d). To fill the actuators with HDSG, we used

a glove box setup illustrated in Supplementary Fig. 15, to prevent contamination with ambient air (see Methods for details).”

Main text, new Figure 6:

“

Figure 6. Use of high-dielectric-strength gas (HDSG) enables actuators with improved performance.

a Schematic of actuators with the same amount of liquid dielectric filled with air (left) and high-dielectric-strength gas (HDSG; right); a C_4F_7N/CO_2 gas mixture was used in this study. Dielectric strength of air is insufficient to prevent dielectric breakdown near the zipping front, while the HDSG provides sufficient insulation with the same amount of liquid dielectric. **b** Force-strain curves of actuators filled with HDSG in the best and worst orientation. The actuation was stable up to 98% HDSG-fill in the best orientation, improved from the 90% threshold with air. In the worst orientation, the actuation was stable up to 80% HDSG-fill, improved from the 70% threshold with air. **c** Comparison between air-filled and HDSG-filled actuators, in the best and worst orientations. **d** Corresponding specific energy of actuators in both best and worst orientations when filled with air or HDSG. In the best orientation, the maximum specific energy reached 51.4 J kg^{-1} at 98% HDSG-fill, substantially higher than the specific energy of 33.5 J kg^{-1} when using air.”

Main text, page 1, line 18 (**Abstract**): “Through theoretical and experimental analyses, we pinpoint the fundamental performance limit as the electrical breakdown in the gas, governed by Paschen's law, thereby providing a guideline for selection of gaseous dielectrics. Using the Peano-HASEL (hydraulically amplified self-healing electrostatic) actuator as a model system, we identify a gas mixture of C_4F_7N and CO_2 that enables outstanding specific energy of 51.4 J kg^{-1} (a nine-fold improvement over conventional Peano-HASELs); using ambient air as gaseous dielectric we still achieve 33.5 J kg^{-1} and a power-to-weight ratio of 1600 W kg^{-1} (a five- and eleven-fold improvement).”

Main text, page 3, line 67 (**Introduction**): “Through a combination of theoretical analysis and experimental validation, we show that Paschen's law can be used to determine the fundamental limits of actuator performance using this strategy, predicting a maximum ratio of gaseous dielectric for a given set of actuation conditions and materials properties of the dielectrics. We used the Peano-HASEL actuator, a type of soft electrohydraulic actuator that linearly contracts upon actuation, as a model system to study the fundamental mechanisms and to demonstrate the benefits of using the solid-liquid-gas architecture. The resulting three-phase soft electrostatic actuators achieved outstanding performance metrics when ambient air was used as the gaseous dielectric—a specific energy of 33.5 J kg^{-1} and a power-to-weight ratio of 1600 W kg^{-1} , which translate to a 532% and 1130% improvement, respectively, compared to the conventional Peano-HASEL actuators. Additionally, the three-phase actuators achieved an 83% higher strain rate (2840 \% s^{-1}) and a 70% wider bandwidth (c.a. 100 Hz) indicating a substantial improvement of actuation speed. We showcased the outstanding performance of these actuators in a jumping robot, which exhibited a 60% increase in jump height and a 32% reduction in take-off time when compared to its solid-liquid counterpart. We also identified a gas mixture of C_4F_7N and CO_2 with particularly high dielectric strength, that enabled outstanding specific energy of 51.4 J kg^{-1} (a nine-fold improvement over

conventional Peano-HASELs), underlining a key advantage of implementing the proposed solid-liquid-gas architectures in closed structures that can enclose gaseous dielectrics with tailored properties.”

Main text, page 13, line 326 (**Discussion**): “We introduced ultralight soft electrostatic actuators based on a solid-liquid-gas architecture, demonstrating substantial performance improvements using Peano-HASELs as a model system. When filled with ambient air as the gaseous dielectric, the three-phase Peano-HASELs achieved a specific energy of 33.5 J kg^{-1} , narrowing the gap to the 40 J kg^{-1} of biological muscles, and a specific power of 1600 W kg^{-1} , which far exceeds the 200 W kg^{-1} typical of muscles^{57,58}. This performance can be further enhanced by employing gases with tailored dielectric properties. Specifically, by using a gas mixture of $\text{C}_4\text{F}_7\text{N}$ and CO_2 , which has a higher dielectric strength than air, the actuators reached a specific energy of 51.4 J kg^{-1} —surpassing that of biological muscle.”

Main text, page 14, line 349 (**Discussion**): “In closed actuators, the gaseous dielectrics can be actively engineered. Here, we have demonstrated that high-dielectric-strength gases can substantially improve actuator stability and performance.”

New Supplementary Figure 15:

“

Supplementary Figure 15. Glove box setup for fabrication of HDSG-filled actuators.

Schematic of the glove box setup used to fill actuators with HDSG, a gas mixture of $\text{C}_4\text{F}_7\text{N}$ and CO_2 . The glove box is maintained at 1 bar CO_2 . A dedicated gas line delivers HDSG into syringes equipped with Luer-lock connectors inside the glove box. Given the 80% CO_2 content of the HDSG, a CO_2 environment in the glove box helps prevent contamination with ambient air.”

Main text, page 16, line 393 (**Methods**): “We filled the actuators with the high-dielectric-strength gas (C₄-FN gas mixture: 20.00%-C₄-FN; 2,3,3,3-tetrafluoro-2-(trifluoromethyl)propanonitrile and 80.00 %-CO₂; carbon dioxide; DILO Amaturen und Anlagen GmbH) in a glove box setup (GB0004; MSE Supplies®) to prevent contamination with ambient air (Supplementary Fig. 15).”

Reviewer 1, comment 3:

“Have the authors verified experimentally that the simplified mathematical model is valid? What about static models in particular?”

Response:

We validated our static model through two independent approaches:

- (a) Theoretical convergence: Our model reduces to a well-established electrostatic actuation model for Peano-HASEL actuators [R1] when the amount of gas-fill approaches 0%, confirming its consistency with existing frameworks.
- (b) Experimental validation: We experimentally verified our static model, particularly the prediction of our model that the relative difference in performance is negligible across varying gas-fill ratios.

[R1] N. Kellaris et al., *An analytical model for the design of Peano-HASEL actuators with drastically improved performance*, Extreme Mech. Lett. 29, 100449, 2019

We added these findings in a new Supplementary Figure 4 as shown below.

New Supplementary Figure 4:

“

Supplementary Figure 4. Model verification.

a Force-strain curves predicted by the new model incorporating compressibility of gaseous dielectric. The new model converges to the established model for Peano-HASEL actuators¹ (red line) when the amount

of gas-fill approaches 0% (black line). **b** Force-strain curves obtained from experiments. As predicted by the model, negligible deviation is observed across gas-fill ratios.”

Reviewer 1, comment 4:

“What are the effects of gravity and how much deviation is there between the experiment and the ideal model in Supplementary Figure 2?”

Response:

The deviation between the experiment and the ideal model is now shown in the new SI figure 4b, as described above in the response to Reviewer 1 comment 3.

Gravity influences the distribution of liquid inside the pouch. It causes the liquid dielectric to pool at the bottom of the actuators. As a result, the air-fill ratio up to which actuation was stable depended on the orientation (up to 90% in the best orientation and 70% in the worst). Below these limits, the effect of gravity on actuator performance during quasi-static actuation was negligible, as the experimentally determined force-strain curves were independent of the orientation. To further examine potential gravitational effects, we performed experiments with actuators in the horizontal orientation. The force-strain curves remained consistent up to 90% air-fill (new panel Fig. 2d), and their specific energy was nearly identical to actuators in vertical orientations (Fig. 2e). Based on these findings, we expect actuators up to 90% air-fill to exhibit consistent performance in any orientation between the best and horizontal orientations. We included these findings in the *Quasi-static performance* section and Figure 2d, e in the main text, as shown below. Additionally, we added a discussion of possible strategies to mitigate the orientation dependence on the performance of the actuators in the **Discussion** section.

Main text, page 7, line 164: “To quantify performance in intermediate orientations, we tested actuators positioned horizontally (Fig. 2d). The force-strain responses closely matched those of the best orientation up to 90% air-fill. This result indicates that up to an air-fill of 90% the performance of actuators will be consistent for all orientations between the best and the horizontal orientation.”

Main text, page 16, line 368 (**Discussion**): “Actuator orientation and dynamic effects were found to influence actuator performance by redistributing liquid dielectric within the pouch. This effect may compromise the practicality of these actuators in certain applications where stable liquid retention between the electrodes cannot be guaranteed. Several strategies could mitigate this effect. Selecting solid-liquid-gas combinations with higher surface tension (e.g., surface treatments or oleophilic metamaterials) can improve liquid retention near the pouch edges, counteracting gravitational or inertial forces. In

addition, incorporating porous internal structures such as sponges or gels may help retain liquid in the desired places and prevent local depletion.”

Figure 2 with new panels with corresponding changes to the figure captions:

“

Figure 2. Evaluation of quasi-static performance of Peano-HASEL actuators based on solid-liquid-gas architectures.

a Schematic of actuators with different amounts of liquid dielectric; sufficient liquid near the zipping front prevents dielectric breakdown, while insufficient liquid increases the probability of breakdown of the gaseous dielectric. **b** Force-strain curves of actuators with varying percentages of air-fill in the "best"

orientation (electrodes at the bottom). Performance was stable up to 90% air-fill, then it declined abruptly, completely failing at 100% air-fill. **c** Force-strain curves of actuators with varying percentages of air-fill in the "worst" orientation (electrodes at the top). Performance was stable up to 70% air-fill (lower than the 90% limit observed in the best orientation), then declined abruptly, completely failing at 100% air-fill. **d** Force-strain curves of actuators with varying percentages of air-fill in the horizontal orientation. Performance was stable up to 90% air-fill, then it declined abruptly, completely failing at 100% air-fill. **e** Specific energy as a function of percentage of air-fill for the three orientations, with a peak of 33.5 J kg⁻¹ at 97% air-fill in the best orientation.”

Reviewer 2

Reviewer 2, comment 1:

“This work presents a novel soft electrostatic actuator based on a solid-liquid-gas structure, which demonstrates significant enhancement on the actuation performance of specific energy and power-to-weight ratio. It is interesting and brings some new perspectives to the electrostatic actuators. Some issues need to be addressed before publication.”

“It is assumed in the manuscript that liquid dielectric always exists at the zipping front due to surface tension. Can the authors use microscope photos or other pictures to prove this?”

Response:

We appreciate this positive evaluation of our work. To verify if liquid dielectric remains present at the zipping front during actuation, we recorded videos of actuators equipped with transparent electrodes and a dyed liquid dielectric. For the transparent electrodes, we used conductive polyacrylamide hydrogel in the best and worst orientations, and transparent Pedot:PSS ink for the horizontal orientation. In all cases, the transparent silicone oil was dyed using beta-carotene as a pigment. The footages confirmed that liquid consistently remains at the zipping front throughout actuation, regardless of orientation. We included these results in the *Quasi-static performance* section, new Supplementary Figure 5, and in the Supplementary Movie 1 (00:41-00:49).

Main text, page 7, line 153: “In this worst orientation, gravity pulls the liquid dielectric away from the zipping front, leaving only a small amount held in place by surface tension, for protection against dielectric breakdown. **Still, the liquid dielectric consistently remains at the zipping front throughout actuation, regardless of the actuator orientation, as shown in Supplementary Fig. 5.**”

Revised Supplementary Movie 1 (00:41-00:49) with additional comment “**Liquid consistently remains at the zipping front**”:

New Supplementary Figure 5:

“

Supplementary Figure 5. Retention of liquid dielectric at zipping front.

Video snapshots demonstrating the presence of liquid at the zipping front throughout actuation, in the best, worst, and horizontal orientations. Best and worst orientation experiments were performed at an earlier stage of research, using hydrogel electrodes; horizontal orientation experiments were performed later using pedot:PSS electrodes—these different experimental conditions should, however, not influence the presence of liquid at the zipping front.”

Reviewer 2, comment 2:

“Will the maximum ratio of gaseous dielectric be influenced by the actuator orientation? If so, the limits should be discussed under the condition of worst orientation.”

Response:

The reviewer is correct that the maximum amount of air-fill depends on the orientation of the actuators as the amount of liquid in the zipping front also depends on the orientation of the actuator. To more clearly differentiate the limits of different orientations, we modified the section *Quasi-static performance* as shown below. We also added an in-depth discussion of the influence of orientation on the actuation performance in the following response to Reviewer 2 comment 3.

Main text, page 7, line 171: “Notably, in the best actuator orientation, specific energy reached a maximum of 33.5 J kg⁻¹ at 97% air-fill, representing a 532% enhancement over a reference actuator with 0% air-fill. In the worst actuator orientation, the specific energy reached a maximum of 23.3 J kg⁻¹ at 95% air-fill, representing a 291% enhancement over a reference actuator with 0% air-fill in the same orientation.”

Reviewer 2, comment 3:

“The effect of actuator orientations on actuation force and strain needs to be further discussed.”

Response:

To further investigate the effect of actuator orientations on actuation force and strain, we performed additional experiments with actuators in the horizontal orientation.

[Please refer to our detailed response below, reproduced from our response to Reviewer 1, comment 4.]

The force-strain curves remained consistent up to 90% air-fill (new panel Fig. 2d), and their specific energy was nearly identical to actuators in vertical orientations (Fig. 2e). Based on these findings, we expect actuators up to 90% air-fill to exhibit consistent performance in any orientation between the best and horizontal orientations. We included these findings in the *Quasi-static performance* section and Figure 2d, e in the main text, as shown below. Additionally, we added a discussion of possible strategies to mitigate the orientation dependence on the performance of the actuators in the **Discussion** section.

Main text, page 7, line 164: “To quantify performance in intermediate orientations, we tested actuators positioned horizontally (Fig. 2d). The force-strain responses closely matched those of the best orientation up to 90% air-fill. This result indicates that up to an air-fill of 90% the performance of actuators will be consistent for all orientations between the best and the horizontal orientation.”

Main text, page 16, line 368 (**Discussion**): “Actuator orientation and dynamic effects were found to influence actuator performance by redistributing liquid dielectric within the pouch. This effect may compromise the practicality of these actuators in certain applications where stable liquid retention between the electrodes cannot be guaranteed. Several strategies could mitigate this effect. Selecting solid-liquid-gas combinations with higher surface tension (e.g., surface treatments or oleophilic metamaterials) can improve liquid retention near the pouch edges, counteracting gravitational or inertial forces. In addition, incorporating porous internal structures such as sponges or gels may help retain liquid in the desired places and prevent local depletion.”

Figure 2 with new panels with corresponding changes to the figure captions:

“

Figure 2. Evaluation of quasi-static performance of Peano-HASEL actuators based on solid-liquid-gas architectures.

a Schematic of actuators with different amounts of liquid dielectric; sufficient liquid near the zipping front prevents dielectric breakdown, while insufficient liquid increases the probability of breakdown of the gaseous dielectric. **b** Force-strain curves of actuators with varying percentages of air-fill in the "best" orientation (electrodes at the bottom). Performance was stable up to 90% air-fill, then it declined abruptly, completely failing at 100% air-fill. **c** Force-strain curves of actuators with varying percentages of air-fill in the "worst" orientation (electrodes at the top). Performance was stable up to 70% air-fill (lower than the

90% limit observed in the best orientation), then declined abruptly, completely failing at 100% air-fill. **d**
Force-strain curves of actuators with varying percentages of air-fill in the horizontal orientation.
Performance was stable up to 90% air-fill, then it declined abruptly, completely failing at 100% air-fill. **e**
Specific energy as a function of percentage of air-fill for **the three** orientations, with a peak of 33.5 **J kg⁻¹**
at 97% air-fill in the best orientation.”

Reviewer 2, comment 4:

“Does the composition or dielectric properties of the air have any impact on actuation behavior?”

Response:

Thank you for your insightful suggestion. Building upon your suggestion, we explored gases with dielectric properties distinct from air, and demonstrated that the use of high-dielectric-strength gas (HDSG) can substantially improve actuator performance.

[Please refer to our detailed response below, reproduced from our response to Reviewer 1, comment 2]

Our model suggests that using high-dielectric-strength gas (HDSG) can lead to considerable performance gains, and thus motivated us to invest substantial experimental efforts to develop a fabrication procedure for HDSG-filled actuators that promise to achieve substantially improved actuator performance. We are excited to report that these efforts have resulted in a substantial improvement of actuator performance: from 33.5 J kg⁻¹ for air-filled actuators to 51.4 J kg⁻¹ for HDSG-filled ones, even surpassing the performance of mammalian muscle (c.a. 40 J kg⁻¹) [R1]. For these experiments, we used a C₄F₇N/CO₂ gas mixture as the HDSG, which offers dielectric strength comparable to SF₆ but with substantially lower global warming potential [R2, R3]. During quasi-static actuation, the HDSG-filled actuators enabled reliable actuation up to 98% fill in the best orientation and up to 80% fill in the worst orientation—considerable improvements over the 90% and 70% thresholds observed with air, respectively.

We added these results as a new Figure 6, which shows a direct performance comparison between air-filled and HDSG-filled actuators, to the manuscript. We also added a new section *High-dielectric-strength gas for enhanced performance in solid-liquid-gas architectures* and revised **Abstract**, **Introduction**, **Discussion**, and **Methods** to implement the new points and findings as shown below.

[R1] J. D. Madden et al., *Artificial muscle technology: physical principles and naval prospects*, IEEE J. Ocean. Eng. 29, 706-728, 2004

[R2] M. Rabie et al., *Assessment of eco-friendly gases for electrical insulation to replace the most potent industrial greenhouse gas SF₆*, Environmental Science & Technology, 2018.

[R3] J. Owens et al. *Recent development of two alternative gases to SF₆ for high voltage electrical power applications*, *Energies*, 2021.

Main text, page 13, line 310:

“*High-dielectric-strength gas for enhanced performance in solid-liquid-gas architectures*

As demonstrated above, a limitation of the performance of electrostatic actuators with solid-liquid-gas architectures, is the dielectric breakdown of the gaseous phase. Closed electrostatic actuators, such as HASELs, HAXELs, and EBMs, offer the possibility of utilizing a high-dielectric-strength gas (HDSG) to improve performance considerably; HDSGs can prevent dielectric breakdown under conditions where air would typically fail (Fig. 6a). To demonstrate this idea, we used a HDSG in the form of a mixture of C₄F₇N and CO₂ (see Methods for details), which offers dielectric strength comparable to SF₆ but with substantially lower global warming potential^{55,56}. With the method used for evaluating quasi-static performance as in Fig. 2 (see Methods for details), we showed that HDSG-filled actuators achieved stable operation up to 98% gas-fill in the best orientation and up to 80% in the worst orientation (Fig. 6b)—considerable improvements over the 90% and 70% thresholds observed with air, respectively. Fig. 6c provides a direct comparison of the force-strain curves of air-filled and HDSG-filled actuators. As a result, the maximum specific energy increased to 51.4 J kg⁻¹ at 98% HDSG-fill in the best orientation, and 30.0 J kg⁻¹ at 95% HDSG-fill in the worst orientation (Fig. 6d). To fill the actuators with HDSG, we used a glove box setup illustrated in Supplementary Fig. 15, to prevent contamination with ambient air (see Methods for details).”

Main text, new Figure 6:

“

Figure 6. Use of high-dielectric-strength gas (HDSG) enables actuators with improved performance.

a Schematic of actuators with the same amount of liquid dielectric filled with air (left) and high-dielectric-strength gas (HDSG; right); a C_4F_7N/CO_2 gas mixture was used in this study. Dielectric

strength of air is insufficient to prevent dielectric breakdown near the zipping front, while the HDSG provides sufficient insulation with the same amount of liquid dielectric. **b** Force-strain curves of actuators filled with HDSG in the best and worst orientation. The actuation was stable up to 98% HDSG-fill in the best orientation, improved from the 90% threshold with air. In the worst orientation, the actuation was stable up to 80% HDSG-fill, improved from the 70% threshold with air. **c** Comparison between air-filled and HDSG-filled actuators, in the best and worst orientations. **d** Corresponding specific energy of actuators in both best and worst orientations when filled with air or HDSG. In the best orientation, the maximum specific energy reached 51.4 J kg^{-1} at 98% HDSG-fill, substantially higher than the specific energy of 33.5 J kg^{-1} when using air.”

Main text, page 1, line 18 (**Abstract**): “Through theoretical and experimental analyses, we pinpoint the fundamental performance limit as the electrical breakdown in the gas, governed by Paschen's law, thereby providing a guideline for selection of gaseous dielectrics. Using the Peano-HASEL (hydraulically amplified self-healing electrostatic) actuator as a model system, we identify a gas mixture of $\text{C}_4\text{F}_7\text{N}$ and CO_2 that enables outstanding specific energy of 51.4 J kg^{-1} (a nine-fold improvement over conventional Peano-HASELs); using ambient air as gaseous dielectric we still achieve 33.5 J kg^{-1} and a power-to-weight ratio of 1600 W kg^{-1} (a five- and eleven-fold improvement).”

Main text, page 3, line 67 (**Introduction**): “Through a combination of theoretical analysis and experimental validation, we show that Paschen’s law can be used to determine the fundamental limits of actuator performance using this strategy, predicting a maximum ratio of gaseous dielectric for a given set of actuation conditions and materials properties of the dielectrics. We used the Peano-HASEL actuator, a type of soft electrohydraulic actuator that linearly contracts upon actuation, as a model system to study the fundamental mechanisms and to demonstrate the benefits of using the solid-liquid-gas architecture. The resulting three-phase soft electrostatic actuators achieved outstanding performance metrics when ambient air was used as the gaseous dielectric—a specific energy of 33.5 J kg^{-1} and a power-to-weight ratio of 1600 W kg^{-1} , which translate to a 532% and 1130% improvement, respectively, compared to the conventional Peano-HASEL actuators. Additionally, the three-phase actuators achieved an 83% higher strain rate (2840 \% s^{-1}) and a 70% wider bandwidth (c.a. 100 Hz) indicating a substantial improvement of actuation speed. We showcased the outstanding performance of these actuators in a jumping robot, which exhibited a 60% increase in jump height and a 32% reduction in take-off time when compared to its solid-liquid counterpart. We also identified a gas mixture of $\text{C}_4\text{F}_7\text{N}$ and CO_2 with particularly high dielectric strength, that enabled outstanding specific energy of 51.4 J kg^{-1} (a nine-fold improvement over conventional Peano-HASELs), underlining a key advantage of implementing the proposed solid-liquid-gas architectures in closed structures that can enclose gaseous dielectrics with tailored properties.”

Main text, page 13, line 326 (**Discussion**): “We introduced ultralight soft electrostatic actuators based on a solid-liquid-gas architecture, demonstrating substantial performance improvements using Peano-HASELs as a model system. When filled with ambient air as the gaseous dielectric, the three-phase Peano-HASELs achieved a specific energy of 33.5 J kg⁻¹, narrowing the gap to the 40 J kg⁻¹ of biological muscles, and a specific power of 1600 W kg⁻¹, which far exceeds the 200 W kg⁻¹ typical of muscles^{57,58}. This performance can be further enhanced by employing gases with tailored dielectric properties. Specifically, by using a gas mixture of C₄F₇N and CO₂, which has a higher dielectric strength than air, the actuators reached a specific energy of 51.4 J kg⁻¹—surpassing that of biological muscle.”

Main text, page 14, line 349 (**Discussion**): “In closed actuators, the gaseous dielectrics can be actively engineered. Here, we have demonstrated that high-dielectric-strength gases can substantially improve actuator stability and performance.”

New Supplementary Figure 15:

“

Supplementary Figure 15. Glove box setup for fabrication of HDSG-filled actuators.

Schematic of the glove box setup used to fill actuators with HDSG, a gas mixture of C₄F₇N and CO₂. The glove box is maintained at 1 bar CO₂. A dedicated gas line delivers HDSG into syringes equipped with Luer-lock connectors inside the glove box. Given the 80% CO₂ content of the HDSG, a CO₂ environment in the glove box helps prevent contamination with ambient air.”

Main text, page 16, line 393 (**Methods**): “We filled the actuators with the high-dielectric-strength gas (C₄-FN gas mixture: 20.00%-C₄-FN; 2,3,3,3-tetrafluoro-2-(trifluoromethyl)propanonitrile and 80.00 %-

CO₂; carbon dioxide; DILO Amaturen und Anlagen GmbH) in a glove box setup (GB0004; MSE Supplies[®]) to prevent contamination with ambient air (Supplementary Fig. 15).”

Reviewer 2, comment 5:

“Long-term stability and reliability are critical for flexible actuators. Have the authors carried out cycle tests?”

Response:

We performed a 100,000-cycle-long durability test on actuators with 0% air (reference) and 70% air in both the best and worst orientations. The actuators were fabricated identically to those used in the frequency response experiments (see *Method*). We applied a sinusoidal excitation voltage (6 kV peak, single polarity) at 5 Hz and approximately 1 N of preload with springs, analogous to the frequency response tests (Supplementary Figure 12, new Supplementary Figure 6b).

For each condition, we tested seven samples. No sample failed within the tested **100,000 cycles**. Stroke measurements from one representative sample per condition were recorded using a laser displacement sensor. The stroke amplitudes, normalized by their respective initial values, remained stable up to 10,000 cycles, after which a gradual decline was observed, likely due to charge retention effects [R1].

Importantly, the level of degradation was similar for all conditions, including the 0% air reference, indicating that the presence of air has negligible impact on long-term durability under these actuation conditions.

[R1] I.-D. Sirbu, et al., *Electrostatic actuators with constant force at low power loss using matched dielectrics*, Nature Electronics, 2023.

These results are added in the section *Dynamic performance*, in a new Supplementary Figure 6b, and Supplementary Note as follows:

Main text, page 12, line 290: “We assessed the long-term durability of the actuators through 100,000-cycle tests. All samples remained functional after 100,000 cycles, with a slight, gradual decline in stroke observed after 10,000 cycles, regardless of air fill or actuator orientation, indicating a negligible impact of air on long-term durability (see Supplementary Fig. 6b and Supplementary Note).”

Supplementary Information page 15, line 238:

“*Prolonged actuation in cyclic actuation*

To assess long-term durability of the actuators, we performed 100,000-cycle-long durability tests with actuators with 0% air (reference) and 70% air in the best and worst orientations (7 samples for each fill

and orientation). The fabrication procedure was identical to those described for the frequency response experiments specified in Methods section in the main text. A sinusoidal voltage signal (6kV peak, single polarity) at 5 Hz was used for actuation (Supplementary Fig. 6b). A nearly constant force of 1 N was applied using the springs specified in Supplementary Fig. 12. The spring resonance frequency was well separated from the 5 Hz actuation frequency.

Under this operating condition, all seven tested samples for each condition—0% air, 70% air in best and worst orientations—remained functional after 100,000 cycles. For each condition, stroke measurements were obtained from one representative sample, using the laser displacement sensor described in the Methods section. The stroke amplitudes, normalized by their respective initial values, remained stable up to 10,000 cycles, after which a gradual decline was observed, likely due to charge retention effects⁶. The level of degradation was similar for all cases, including the reference actuator with 0% air, indicating that the presence of air has negligible impact on long-term durability under these actuation conditions.”

New Supplementary Figure 6b and corresponding figure caption:

“

Supplementary Figure 6. Prolonged actuation at constant voltage and in cyclic actuation.

b Normalized amplitude over 100,000 cycles under a sinusoidal voltage signal, for actuators with 0% air-fill (reference) and with 70% air-fill. At this specific actuation condition, all seven samples for each case remained functional after 100,000 cycles. Actuator stroke was normalized with the initial stroke of each actuator, to show the actuator behavior over many cycles. For all three cases, the normalized amplitude remained stable up to 10,000 cycles, after which a gradual decline was observed, likely due to charge

retention effects⁶. The level of degradation was similar for all cases, including the reference actuator with 0% air, suggesting that the presence of air has negligible impact on long-term durability.”

Reviewer 2, comment 6:

“Since the liquid can move and oscillate in the air, the three-phase actuators are seem to be more unstable compared to the conventional Peano-HASELs. How to ensure the stability of the actuator output, especially in periodic high-frequency response?”

Response:

This effect was only noticeable at frequencies above 10 Hz in our experiments (Figure 4g). At higher actuation frequencies, the data indeed suggest that redistribution of the liquid dielectric within the pouch can affect performance. To address this instability of the actuation in high frequencies, we now propose strategies to mitigate redistribution of liquid in the **Discussion** section, including the use of solid-liquid-gas combinations with higher surface tension and surface treatments of solid dielectric, to enhance liquid retention:

[Please refer to our detailed response below, reproduced from our response to Reviewer 2, comment 3.]

Main text, page 16, line 368 (**Discussion**): “Actuator orientation and dynamic effects were found to influence actuator performance by redistributing liquid dielectric within the pouch. This effect may compromise the practicality of these actuators in certain applications where stable liquid retention between the electrodes cannot be guaranteed. Several strategies could mitigate this effect. Selecting solid-liquid-gas combinations with higher surface tension (e.g., surface treatments or oleophilic metamaterials) can improve liquid retention near the pouch edges, counteracting gravitational or inertial forces. In addition, incorporating porous internal structures such as sponges or gels may help retain liquid in the desired places and prevent local depletion.”

Reviewer 3

Reviewer 3, general comment:

“The manuscript presents an interesting approach to modifying the well-known Peano-HASEL actuator by partially replacing the liquid dielectric with a gas, forming a solid–liquid–gas architecture. The authors provide high quality and extensive experimental validation, including quasi-static tests, dynamic response analyses, and a jumping robot demonstration. The proposed idea reduces mass and provide important improvements in performance metrics (specific energy, strain rate, power-to-weight ratio) of the Peano actuator; however, the conceptual novelty, solely connected to the partial filling approach, appears limited since the fundamental actuator design is already established in previous work (adequately cited in the manuscript).”

Response:

We appreciate the positive evaluation of the quality of our work; regarding conceptual novelty, we would like to refer to our response to reviewer 1, general comment:

[Please refer to our detailed response below, reproduced from our response to Reviewer 1, general comment.]

As the reviewer points out, the performance improvement enabled by the solid-liquid-gas architecture is one important outcome of this paper. Additionally, the paper features two novel aspects, which we feel are important advances over existing literature; we have revised the manuscript to more clearly articulate these advances:

- 1) HASEL actuators and the actuators developed by Rossiter and colleagues share liquid-enhanced electrostatic zipping as an actuation mechanism. However, Rossiter’s actuators are open structures, which directly interact with the surrounding air, whereas HASELs based on the solid-liquid-gas architecture are closed structures, where air can be replaced with other gases that feature high dielectric strength. Stimulated by insightful comments of reviewers 1 and 2, in the process of this major revision, we have identified a C_4F_7N/CO_2 gas mixture for use as high-dielectric-strength gas (HDSG); we have further developed a fabrication procedure for HDSG-filled actuators that achieve substantially improved actuator performance (33.5 J kg^{-1} for air-filled actuators $\rightarrow 51.4 \text{ J kg}^{-1}$ for HDSG-filled actuators; see detailed discussion in **Reviewer 1, comment 2**).
- 2) Building upon Paschen’s law, we derived a model to predict dielectric breakdown through gas in HASELs based on the solid-liquid-gas architecture; through theoretical and experimental analyses, we pinpoint the fundamental performance limit as the electrical breakdown in the gas,

thereby providing a guideline for selection of gaseous dielectrics. The model introduced in the manuscript should, in principle, also be applicable for determining the minimum amount of liquid in Rossiter's actuators. To confirm that the model also works for open structures, we performed dielectric breakdown tests of liquid-gas interfaces in a simplified setup (see detailed discussion in **Reviewer 3, comment 1**)

Reviewer 3, comment 1:

“An accompanying theoretical framework based on Paschen's law is introduced (which could potentially represent a valuable contribution) to explain the onset of dielectric breakdown, but there are some critical points regarding this. For a convincing validation, I suggest performing experiments on a dedicated, simplified, controlled setup aimed at isolating and validating the model parameters. This would help to limit the number of uncontrolled variables.”

Response:

We thank the reviewer for this constructive and important suggestion. As suggested, we conducted dielectric breakdown tests of liquid-gas interfaces in a simplified setup, which demonstrated a good alignment with our theoretical framework based on Paschen's law.

As the reviewer noted, in the main manuscript, the air pressure inside actuator P and electrode gap d were inferred from analytical models rather than directly measured. To validate our theoretical framework under more controlled conditions, we designed a simplified experimental setup using rigid acrylic plates that were angled with respect to each other to replicate the zipping front of the actuators, as shown in a new Supplementary Figure 9a and b.

In this setup, carbon electrodes were printed on the outer side of solid dielectric films which were securely bonded to the acrylic plates to prevent movement during voltage application. The electrode gap was filled with varying amounts of liquid dielectric—sufficient, insufficient, and none (Supplementary Fig. 9c). In this setup open to the ambient air, the air pressure P was constant at atmospheric pressure, and the electrode gap d remained constant.

We applied the activation voltage pattern used in the tests described in the *Identification of fundamental performance limits* section in the main text: a ramp of 1 kV s^{-1} with a 2-second hold at each 1 kV step from 5 to 10 kV, repeated for 100 cycles with alternating polarity. With the application of the voltage, as shown in the new Supplementary Figure 9d, the setup with sufficient liquid dielectric (left, electrodes fully covered) exhibited only a few and minor current spikes, indicating effective insulation. In contrast, the setup with insufficient liquid dielectric (middle; electrodes partially covered) exhibited frequent

current spikes, particularly at higher voltages. The setup with no liquid dielectric (right) exhibited numerous current spikes, even at low voltages. The new Supplementary Figure 9e provides an overview of the distribution of voltages at which the first current spikes occurred in each cycle.

To quantitatively test the theoretical prediction by Paschen's law, we analyzed the case of insufficient-liquid. The atmospheric pressure was measured to be 973.3 hPa (730.0 Torr) using a barometer, and the electrode gap at the liquid-air interface was estimated to be 1.70 mm based on image analysis. These values were then plotted on the Paschen's curve for air (new Supplementary Fig. 9f). The intersection point of the orange (Paschen's curve for air) and black (measured value) lines indicates the predicted impending dielectric breakdown of air within the setup. The zoomed-in view shows that most breakdown events align well with the value predicted from Paschen's law.

We have added these findings as a new Supplementary Figure 9, and updated the main text in the *Identification of fundamental performance limits* section as follows:

Main text, page 10, line 236: “This hypothesis was further confirmed with a simplified experimental setup with measured p and d in an open system (Supplementary Fig. 9; see Methods for details).”

Main text, page 17, line 410 (**Methods**):

“*Fabrication and experimental details of the dielectric breakdown test in a simplified setup*

Rigid acrylic plates were fixed at an angle of 27.6° relative to each other. Biaxially oriented polyester films (Mylar 850-15, Petroplast GmbH), each bearing 2 cm x 6 cm carbon electrodes (CI-2051, Nagase ChemteX America LLC) printed on their outer surfaces, were securely bonded to the angled acrylic plates. The sides of the assembly were sealed using thin transparent acrylic plates. A varying amount of liquid dielectric (Silicone oil M5, Carl Roth) was added to fill the electrode gap. A voltage ramp was applied at a rate of 1 kV s^{-1} , with a 2-second hold at each 1 kV increment from 5 to 10 kV. This pattern was repeated for 100 cycles, alternating polarity each cycle. Using the measured current profiles, the breakdown voltages were recorded as the voltages at which the first current spikes occurred in each cycle.”

New Supplementary Figure 9:

“

Supplementary Figure 9. Dielectric breakdown test of liquid-gas interfaces in a simplified setup.

A simplified experimental setup using rigid acrylic plates (a, b) to replicate the zipping front of actuators under three conditions (c): sufficient (left), insufficient (middle), and no (right) liquid dielectric. Carbon electrodes were printed on the outer surfaces of solid dielectric films which were securely bonded to the acrylic plates. d Measured current profiles for each condition, when actuators were exposed to the typical activation voltage pattern used for actuator evaluation. Therefore, an identical voltage signal ramping at 1

kV s⁻¹ and pausing for 2 seconds at every 1 kV step from 5 to 10 kV was applied for 100 cycles (10 voltage cycles are shown in the figure; however, current data for all 100 cycles is shown, so that all current spikes are visible), with polarity reversed each cycle. The setup with sufficient liquid dielectric (left; electrodes fully covered) exhibited only a few and minor current spikes, indicating effective insulation. In contrast, the setup with insufficient liquid dielectric (middle; electrodes partially covered) exhibited frequent current spikes, particularly at higher voltages. The setup with no liquid dielectric (right) exhibited numerous current spikes, even at low voltages. **e** Voltages at which the first current spikes occurred in each cycle. The white line denotes the median value, and the thick black bars denote the standard deviation; the thin black bar shows the data range from lowest to highest value, avoiding outliers. The width of the colored areas indicates the number of recorded breakdown events for a given voltage. **f** Paschen's curve for air (orange) intersecting with a black line denoting the measured value of atmospheric pressure p , which was measured to be 973.3 hPa (730.0 Torr), and the electrode gap along the liquid-air interface d , which was 1.70 mm. The intersection point of the orange and black lines indicates the predicted impending dielectric breakdown of air within the setup. The zoomed-in view shows that most breakdown events align well with the value predicted from Paschen's law."

Reviewer 3, comment 2:

"Although introducing air reduces mass, the inherent compressibility of gas could lead to unwanted effects when the actuator is subjected to temperature and pressure variations. The authors should address whether changes in these conditions might alter performance (e.g., stroke). A simple dedicated theoretical/experimental study examining the influence of thermal/pressure expansion on actuation behavior should be added. A discussion should also be introduced, highlighting possible connected limitations (e.g., underwater robotics?)"

Response:

We expanded our analytical model to incorporate the effects of external pressure and temperature. This updated model allows us to predict performance of actuator (in terms of force-strain behavior) under various environmental conditions.

Assuming fabrication under standard room conditions ($p_0 = 1.0$ bar, $T_0 = 20$ °C) and isothermal actuation under different external pressure and temperature conditions, we examined deviations under design conditions enveloping representative outdoor conditions. We varied external pressure ($p_{\text{ext}} = 0.8$ to 1.1 bar) at the same external temperature ($T_{\text{ext}} = 20$ °C), and the external temperature ($T_{\text{ext}} = -30$ to 70 °C) at the same external pressure ($p_{\text{ext}} = 1.0$ bar), and compared the resulting force-strain curves. Here, we present only the 100% gas-filled actuator case as it is the most extreme case regarding environmental

effects. Even though the actuator does not work in experiments with 100% gas-fill (due to dielectric breakdown through the gas gap), in our model it allows a conservative estimation of the influence of the compressibility of the gas on the actuation performance, when the external pressure and temperature change.

The new Supplementary Figure 16a and b show that changes in external pressure and temperature change both the maximum generated actuation forces and the maximum actuation strain, as the contained gas changes volume. These variations remain small for typical variations of the environmental conditions.

As the reviewer noted, in extreme environments, compressibility effects become more pronounced. However, it is possible to counteract changes in actuation behavior through fabrication and design changes, as we show with an example that could occur, in extreme environments such as underwater. As shown in the new Supplementary Figure 16c, increasing the external pressure to 3.0 bar at an external temperature of 20 °C considerably changes the force-strain curve of an actuator that was filled with gas $p_0 = 1.0$ bar and $T_0 = 20$ °C. The resulting decrease of the gas volume increases the blocking force, but drastically reduces the maximum actuation stroke, because the electrodes are fully zipped before the angle α reaches 90 °. By increasing the electrode length, the full range of actuation strains can be achieved (dashed line in Supplementary Fig. 16c). Additionally, when a high external pressure is anticipated, the actuator can be pre-filled at a higher pressure to match the external pressure ($p_{\text{fill}} = 3.0$ bar to match the $p_{\text{ext}} = 3.0$ bar); compressibility effects then become negligible, and the original performance is recovered. Of course, when the external pressure conditions undergo dynamic changes (such as when an underwater robot dives up and down), the force-strain characteristics of actuators would also undergo dynamic changes; in these cases, the control algorithms for the actuators would have to be adapted to account for these changes in actuation behavior.

We added these results in the Supplementary Information, revised Supplementary Figure 2, new Supplementary Figure 16, and **Discussion**.

Supplementary Information page 6, line 81:

A. Isothermal actuation

During isothermal actuation, the actuator deforms sufficiently slowly to always be thermally equilibrated to the surroundings (at constant external pressure p_{ext} and temperature T_{ext}).

Assuming ideal gas behavior, the equation of the state of the contained gas can be described as

$$\frac{pV_{\text{gas}}}{T} = \frac{p_{\text{gas},0}V_{\text{gas},0}}{T_0} \quad (20)$$

Combining equations (13), (18), and (20) leads to

$$p = p_{\text{ext}} + \frac{F}{w(L_p - l_e)} \left(\frac{\alpha}{\cos \alpha} \right) = p_0 \frac{V_{\text{gas},0}}{\frac{w(L_p - l_e)^2}{2} \left(\frac{\alpha - \sin \alpha \cos \alpha}{\alpha^2} \right) - V_{\text{liq}}} \frac{T_{\text{ext}}}{T_0} \quad (21)$$

B. Isentropic actuation

During isentropic actuation, the actuator deforms sufficiently fast to prevent any heat exchange with the surroundings. Assuming ideal gas behavior, the equation of the state of the contained gas can be described as

$$pV_{\text{gas}}^\gamma = p_0V_{\text{gas},0}^\gamma \quad (\text{where } \gamma = \frac{c_p}{c_v}) \quad (22)$$

Combining equations (13), (18), and (22) leads to

$$p = p_{\text{ext}} + \frac{F}{w(L_p - l_e)} \left(\frac{\alpha}{\cos \alpha} \right) = p_0 \left(\frac{V_{\text{gas},0}}{\frac{w(L_p - l_e)^2}{2} \left(\frac{\alpha - \sin \alpha \cos \alpha}{\alpha^2} \right) - V_{\text{liq}}} \right)^\gamma \quad (23)$$

With the calculated $\alpha(F, \Phi)$ from equation (19), equations (21) or (23) can be used to calculate l_e as a function of given external load F , voltage Φ , volume of liquid V_{liq} , external pressure p_{ext} , and in case of isothermal actuation, temperature T_{ext} .

$$l_e = g(F, \Phi, V_{\text{liq}}, p_{\text{ext}}, (T_{\text{ext}})) \quad (24)$$

Eventually, equations (19) and (24) uniquely define the two independent geometric variables α and l_e at the given actuation conditions; ($\alpha(F, \Phi)$, $l_e(F, \Phi, V_{\text{liq}}, p_{\text{ext}}, (T_{\text{ext}}))$).

Revised Supplementary Figure 2 and corresponding figure caption:

“

Supplementary Figure 2. Analytical model of three-phase Peano-HASEL actuators.

a Schematic of the actuator at fabricated state, with a given geometry (length L_p , width w , electrode length L_e) and selection of dielectrics (liquid volume V_{liq} , initial gas volume $V_{gas,0}$) under the atmospheric pressure ($p_0 = p_{atm}$) and room temperature ($T_0 = T_{room}$). The angle α parameterizes the shape of the

actuator. **b** Schematic of the actuator at initial state, under different pressure and temperature conditions (p_{ext} and T_{ext}). The angle α parameterizes the shape of the actuator. The volume of the gas inside the actuator will be affected by the external pressure and temperature conditions, thus changing the initial central angle α_0 and initial length of actuator l_0 . **c** Actuator during actuation (applied load F , voltage Φ), with electrodes zipped to a length l_e . The actuation process changes the gas pressure p , and thus the volume of gas V_{gas} . Stored electrical energy is determined by the shell material (thickness t ; dielectric constant ϵ_s).”

New Supplementary Figure 16:

“

Supplementary Figure 16. Effects of external pressure and temperature during isothermal actuation

a Calculated force-strain curves of a 100% gas-filled actuator under different external pressures (0.8 and 1.1 bar) at 20°C , when the actuator is filled at a pressure $p_0 = 1.0$ bar and temperature $T_0 = 20^\circ\text{C}$. **b** Calculated force-strain curves under different external temperatures (-30 and 70°C) at 1.0 bar, when the

actuator is filled at a pressure $p_0 = 1.0$ bar and temperature $T_0 = 20$ °C. c Calculated force-strain curves under 3.0 bar external pressure at 20 °C, representing an underwater condition. When the actuator is initially filled at 1.0 bar, it exhibits reduced strain at low forces and enhanced strain at high forces due to the compressed internal gas volume. Under such high pressure, the gas volume considerably decreases, requiring extended electrodes to reach the cylindrical shape at fully zipped state, especially at low forces. When the actuator is filled at 3.0 bar to match the external pressure, compressibility becomes negligible and the original force-strain behavior is recovered.”

Supplementary Information page 8, line 108: “In Supplementary Fig. 16, the changes in force-strain characteristic under different external temperature and pressure conditions is predicted using 100% gas-filled actuator as the most extreme case regarding environmental effects. Even though the actuator does not work in experiments with 100% gas fill (due to dielectric breakdown through the gas gap), in our model it allows a conservative estimation of the influence of the compressibility of the gas on the actuation performance, when the external pressure and temperature change. Again, materials system and geometry of the actuators are identical to the ones described in Methods section in the main text. Assuming fabrication under standard room conditions ($p_0 = 1.0$ bar, $T_0 = 20$ °C) and isothermal actuation under different external pressure and temperature conditions, we examined deviations under design conditions enveloping representative outdoor conditions. We varied the external pressure ($p_{\text{ext}} = 0.8$ to 1.1 bar) at the same external temperature ($T_{\text{ext}} = 20$ °C), and the external temperature ($T_{\text{ext}} = -30$ to 70 °C) at the same external pressure ($p_{\text{ext}} = 1.0$ bar), and compared the resulting force-strain curves.

Supplementary Fig. 16 a and b show that changes in external pressure and temperature change both the maximum generated actuation forces and the maximum actuation strain, as the contained gas changes volume. These variations remain small for typical variations of the environmental conditions. It is possible to counteract changes in actuation behavior through fabrication and design changes, as we show with an example that could occur, in extreme environments such as underwater. As shown in the Supplementary Fig. 16c, increasing the external pressure to 3.0 bar at an external temperature of 20 °C considerably changes the force-strain curve of an actuator that was filled with gas $p_0 = 1.0$ bar and $T_0 = 20$ °C. The resulting decrease of the gas volume increases the blocking force, but drastically reduces the maximum actuation stroke, because the electrodes are fully zipped before the angle α reaches 90 °. By increasing the electrode length, the full range of actuation strains can be achieved (dashed line in Supplementary Fig. 16c). Additionally, when a high external pressure is anticipated, the actuator can be pre-filled at a higher pressure to match the external pressure ($p_{\text{fill}} = 3.0$ bar to match the $p_{\text{ext}} = 3.0$ bar); compressibility effects then become negligible, and the original force-strain behavior is recovered.”

Main text, page 15, line 355 (**Discussion**): “Our analytical model, which incorporates compressibility of gas, also provides insights into how actuator performance may vary under different environmental pressures and temperatures. Although the compressibility of the gas had negligible effects under standard conditions, performance may differ when the environmental conditions change (Supplementary Fig. 16a and b). Whereas the changes in actuation behavior are small for typical environmental variations, compressibility effects become more pronounced in extreme pressure and temperature environments. For example, high external pressure environments during underwater operation can noticeably compress the contained gas, severely altering the actuation behavior (Supplementary Fig. 16c). In such cases, however, the gas fill and the actuator design can be adjusted—pre-filling actuators at a higher pressure to match the external pressure, and increasing electrode length—to the targeted environmental operating conditions (Supplementary Fig. 16c). When the external pressure conditions undergo dynamic changes (such as when an underwater robot dives up and down), the force-strain characteristics of actuators would also undergo dynamic changes; in these cases, the control algorithms for the actuators would have to be adapted to account for these changes in actuation behavior.”

Reviewer 3, comment 3:

“The cyclical operational stability of the actuator should be shown, as differences may appear between the first and subsequent cycles due to changes induced in the interfaces between the fluid and gas. Thus, it would be valuable to see cyclical data and (possibly) preliminary discussion on degradation mechanisms that might occur over many cycles.”

Response:

[Please refer to our detailed response below, reproduced from our response to Reviewer 2, comment 5.]

We performed a 100,000-cycle-long durability test on actuators with 0% air (reference) and 70% air in both the best and worst orientations. The actuators were fabricated identically to those used in the frequency response experiments (see *Method*). We applied a sinusoidal excitation voltage (6 kV peak, single polarity) at 5 Hz and approximately 1 N of preload with springs, analogous to the frequency response tests (Supplementary Figure 12, new Supplementary Figure 6b).

For each condition, we tested seven samples. No sample failed within the tested **100,000 cycles**. Stroke measurements from one representative sample per condition were recorded using a laser displacement sensor. The stroke amplitudes, normalized by their respective initial values, remained stable up to 10,000 cycles, after which a gradual decline was observed, likely due to charge retention effects [R1].

Importantly, the level of degradation was similar for all conditions, including the 0% air reference,

indicating that the presence of air has negligible impact on long-term durability under these actuation conditions.

[R1] I.-D. Sirbu, et al., *Electrostatic actuators with constant force at low power loss using matched dielectrics*, Nature Electronics, 2023.

These results are added in the section *Dynamic performance*, in a new Supplementary Figure 6b, and Supplementary Note as follows:

Main text, page 12, line 290: “We assessed the long-term durability of the actuators through 100,000-cycle tests. All samples remained functional after 100,000 cycles, with a slight, gradual decline in stroke observed after 10,000 cycles, regardless of air fill or actuator orientation, indicating a negligible impact of air on long-term durability (see Supplementary Fig. 6b and Supplementary Note).”

Supplementary Information page 15, line 238:

“*Prolonged actuation in cyclic actuation*

To assess long-term durability of the actuators, we performed 100,000-cycle-long durability tests with actuators with 0% air (reference) and 70% air in the best and worst orientations (7 samples for each fill and orientation). The fabrication procedure was identical to those described for the frequency response experiments specified in Methods section in the main text. A sinusoidal voltage signal (6kV peak, single polarity) at 5 Hz was used for actuation (Supplementary Fig. 6b). A nearly constant force of 1 N was applied using the springs specified in Supplementary Fig. 12. The spring resonance frequency was well separated from the 5 Hz actuation frequency.

Under this operating condition, all seven tested samples for each condition—0% air, 70% air in best and worst orientations—remained functional after 100,000 cycles. For each condition, stroke measurements were obtained from one representative sample, using the laser displacement sensor described in the Methods section. The stroke amplitudes, normalized by their respective initial values, remained stable up to 10,000 cycles, after which a gradual decline was observed, likely due to charge retention effects⁶. The level of degradation was similar for all cases, including the reference actuator with 0% air, indicating that the presence of air has negligible impact on long-term durability under these actuation conditions.”

New Supplementary Figure 6b and corresponding figure caption:

“

Supplementary Figure 6. Prolonged actuation at constant voltage and in cyclic actuation.

b Normalized amplitude over 100,000 cycles under a sinusoidal voltage signal, for actuators with 0% air-fill (reference) and with 70% air-fill. At this specific actuation condition, all seven samples for each case remained functional after 100,000 cycles. Actuator stroke was normalized with the initial stroke of each actuator, to show the actuator behavior over many cycles. For all three cases, the normalized amplitude remained stable up to 10,000 cycles, after which a gradual decline was observed, likely due to charge retention effects⁶. The level of degradation was similar for all cases, including the reference actuator with 0% air, suggesting that the presence of air has negligible impact on long-term durability.”

Reviewer 3, comment 4:

“The model presented in the supplementary material applies Boyle’s law ($pV = \text{constant}$) under the assumption of isothermal conditions. While this assumption may hold in quasi-static regimes, it should be clarified how dynamic conditions—where heat exchange may occur at a slower rate—could affect the model’s validity.”

Response:

The reviewer is correct that the temperature of the gas may change during non-quasi-static deformation. While isothermal deformation represents perfect thermal equilibrium with the surroundings, isentropic deformation represents the opposite extreme—no heat exchange at all. The actual behavior of actuators will lie between the isothermal and isentropic extremes. We added a model modified for isentropic

deformation to the Supplementary Information. This model leads to nearly the same force-strain curve as isothermal deformation and the reference actuator (0% air), as shown in the new panel Supplementary Fig. 3e. As a result, we conclude that dynamic conditions will lead to the almost same actuation behavior as for isothermal conditions.

We made the following changes to the main text and Supplementary Information:

Main text, page 7, line 146: “The model predicts negligible deviation between isentropic and isothermal extremes under typical actuation conditions (see Supplementary Fig. 3e); therefore, we assume isothermal conditions throughout this study without compromising the validity of the results.”

[Please refer to our derivation below, reproduced from our response to Reviewer 3, comment 2. The resulting Supplementary Figure 3 and Supplementary Note are newly presented here in response to this comment.]

Supplementary Information page 6, line 86:

“

B. Isentropic actuation

During isentropic actuation, the actuator deforms sufficiently fast to prevent any heat exchange with the surroundings. Assuming ideal gas behavior, the equation of the state of the contained gas can be described as

$$pV_{\text{gas}}^{\gamma} = p_0V_{\text{gas},0}^{\gamma} \quad (\text{where } \gamma = \frac{c_p}{c_v}) \quad (22)$$

Combining equations (13), (18), and (22) leads to

$$p = p_{\text{ext}} + \frac{F}{w(L_p - l_e)} \left(\frac{\alpha}{\cos \alpha} \right) = p_0 \left(\frac{V_{\text{gas},0}}{\frac{w(L_p - l_e)^2}{2} \left(\frac{\alpha - \sin \alpha \cos \alpha}{\alpha^2} \right) - V_{\text{liq}}} \right)^{\gamma} \quad (23)$$

With the calculated $\alpha(F, \Phi)$ from equation (19), equations (21) or (23) can be used to calculate l_e as a function of given external load F , voltage Φ , volume of liquid V_{liq} , external pressure p_{ext} , and in case of isothermal actuation, temperature T_{ext} .

$$l_e = g(F, \Phi, V_{\text{liq}}, p_{\text{ext}}, (T_{\text{ext}})) \quad (24)$$

Eventually, equations (19) and (24) uniquely define the two independent geometric variables α and l_e at the given actuation conditions; ($\alpha(F, \Phi)$, $l_e(F, \Phi, V_{liq}, p_{ext}, (T_{ext}))$).

[The results shown below are newly presented]

Supplementary Figure 3 with new panels with corresponding changes to the figure captions:

“

Supplementary Figure 3. Results of the analytical model of three-phase Peano-HASEL actuators.

a Schematic illustration and parameterization of the model. **b** Theoretically calculated volume of pouch and zipped length of the actuators, at 0% (fully liquid-filled) and 100% (fully gas-filled) gas-fill ratios, under 8 kV voltage. The compressibility of the gas leads to pressure and volume changes of the gas during actuation. **c** Pouch shape of actuators at 0% and 100% gas-fill at 8 kV voltage and 20 N force, drawn based on calculations. Despite an increased gas pressure and a resulting reduction in gas volume, the larger zipped length ($l_{e, \text{gas}} > l_{e, \text{liq}}$) compensates for the compressed gas volume, yielding nearly identical actuator lengths ($l_{\text{gas}} \approx l_{\text{liq}}$). **d** Force-strain curves of actuators with varying gas-fills, based on the analytical model, assuming isothermal deformation. **e** Force-strain curves of 100% gas-filled actuators are compared under both isothermal and isentropic processes.”

Supplementary Information page 7, line 105: “Notably, isentropic deformation and isothermal deformation lead to very similar force-strain curves compared to a reference actuator with 0% air-fill. In a real application, the conditions will lie between the isothermal and isentropic extremes, which shows that compressibility is negligible under the analyzed conditions.”

Reviewer 3, comment 5:

“It would be beneficial to know if any force relaxation related to charge accumulation behavior was observed during testing and whether the introduction of gas has any influence on such phenomena.”

Response:

To address this question, we performed actuation tests, during which we measured the actuation strain as a function of time under constant voltage and constant force. We tested three actuators: a reference actuator with 0% air-fill and actuators with 70% air-fill in the best and worst orientations. All actuators maintained a nearly constant amplitude over 600s; the air-filled actuators did not exhibit worse relaxation compared to the reference actuator.

These results are added in the section *Quasi-static performance*, in a new Supplementary Figure 6a, and Supplementary Note as follows:

Main text, page 7, line 161: “We tested the influence of air fill on force relaxation related to charge accumulation⁵². During prolonged actuation under constant voltage and force, the introduction of air had a negligible effect on charge accumulation behavior (Supplementary Note, Supplementary Fig. 6a).”

New Supplementary Figure 6a and corresponding figure caption:

“

Supplementary Figure 6. Prolonged actuation at constant voltage and in cyclic actuation.

a Actuation strokes normalized by their initial strokes as a function of time, under constant voltage and force. For all three cases, the normalized amplitude remained almost constant, indicating no negative influence from the presence of air.”

Supplementary Information page 15, line 227:

“*Prolonged actuation at constant voltage*

To test force relaxation related to charge accumulation⁶, we measured the actuation stroke of actuators under constant voltage and force for 600 seconds. We compared an actuator with 0% air-fill (reference) with actuators with 70% air-fill in the best and worst orientations. At $t = 0$, a constant voltage of 8 kV and a constant force of 1 N were applied, using the high voltage amplifier and the dual-mode lever system described in the Methods section in the main text (also see Supplementary Fig. 1). For all three cases, the normalized amplitude—normalized with the initial stroke of each actuator—remained nearly constant over the duration of 600 seconds. After 600 seconds, the amplitudes of the 0% air-filled actuator (reference) and the 70% air-filled actuator in the best orientation dropped by 3.3% and 0.0004% from their initial amplitudes, respectively, while the 70% air-filled actuator in the worst orientation showed an increase of 1.6%. These findings suggest that the introduction of air has a negligible influence on the charge accumulation behavior.”

Reviewer 1

“The paper maintains its novelty by adding the gas with high dielectric strength in response to the previous comment. However, Figures 3, 4, and 5 remain unchanged. Considering the overall scenario, experiments and considerations regarding gases with high dielectric strength should be added to ensure consistency.”

Response:

Following this constructive reviewer suggestion, we have performed additional experiments to elucidate the influence of the high-dielectric-strength gas (HDSG; C₄F₇N/CO₂ mixture) in the other experiments presented in this study. Figures 3, 4, and 5 include: (i) actuation failures modeled by Paschen’s law (Figure 3), (ii) evaluation of dynamic performance (Figure 4), and (iii) demonstration of enhanced actuator performance in a jumping robot (Figure 5).

Specifically, we performed additional experiments, focusing on dynamic actuator performance. Under sinusoidal voltage input (± 6 kV), HDSG-filled actuators exhibited a broader bandwidth, maintaining their amplitude at higher frequencies where the air-filled actuators exhibited reduced amplitude (i.e., below -3 dB relative to their amplitude at 1 Hz). Under step voltage input (8 kV), however, the actuation threshold remained unchanged: both air-filled and HDSG-filled actuators operated stably up to 90% fill and failed beyond 95% fill, resulting in similar peak specific power profiles. These results indicate that while the higher dielectric strength of HDSG effectively mitigates breakdown during high-frequency cyclic actuation—where gas pressure and electrode gap distance vary with the voltage—it does not fully suppress breakdown under rapid voltage changes where these internal parameters remain constant.

Given that the step-response performance of HDSG-filled actuators showed negligible improvement over air-filled ones, the performance of the jumping robot shown in Figure 5 does not substantially change when replacing air with HDSG.

Figure 3, which focuses on identifying the fundamental performance limits, cannot be replicated using HDSG because no Paschen’s curve data (breakdown voltage across different pressure-distance values) are available in the literature for this gas mixture, and generating such data lies beyond the scope of this paper. Nevertheless, the experiments conducted with air in the previous version sufficiently validated our key hypothesis that actuator failure originates from dielectric breakdown in the gaseous phase.

We have incorporated the new experimental findings into the revised manuscript as Figure 7 within the section *High-dielectric-strength gas for enhanced performance in solid-liquid-gas architectures*, and as Supplementary Figure 17, as shown below:

Main text, new Figure 7:

“

Figure 7. Use of high-dielectric-strength gas (HDSG) enables actuators with improved bandwidth.

Normalized amplitude of the actuators filled with air (dashed lines) and high-dielectric-strength gas (HDSG; solid lines). Actuators filled with the HDSG (C_4F_7N/CO_2 gas mixture) exhibited increased bandwidth compared to the actuators filled with the same amount of air. **a** In the best orientation, actuators filled with 60% HDSG achieved a bandwidth above 100 Hz, improved from 80 Hz of the 60% air-filled actuator. Similarly, 80% and 90% HDSG-filled actuators achieved improved bandwidths of 90 Hz and 70 Hz, compared to 40 Hz and 10 Hz for the corresponding air-filled actuators, respectively. **b** In the worst orientation, actuators filled with 80% and 90% HDSG achieved bandwidths of 80 Hz and 20 Hz, improved from 10 Hz and 2 Hz for the corresponding air-filled actuators, respectively.

”

New Supplementary Figure 17:

“

Supplementary Figure 17. Evaluation of actuator performance under step voltage input with high-dielectric-strength gas (HDSG). Peak strain rate, peak power, and peak specific power of actuators filled with air (marked with crosses) and high-dielectric-strength gas (HDSG; marked with circles), when actuators are driven by a step voltage input. **a** In the best orientation, air-filled actuators exhibited stable performance up to 90% air-fill, following the blue trend line obtained from the gas-dominant group (filled with air) in the main text; actuation failed beyond 90% fill. The use of HDSG did not extend this threshold. **b** In the worst orientation, HDSG-filled actuator showed stable actuation up to 40% fill, which was not improved compared to the 60% of air-filled actuators. Error bars represent standard deviations for $n \geq 5$ trials.

”

Main text, page 14, line 323: “To fill the actuators with HDSG, we used a glove box setup illustrated in Supplementary Fig. 16, to prevent contamination with ambient air (see Methods for details). With the method used for evaluating quasi-static performance as in Fig. 2 (see Methods for details), we showed that HDSG-filled actuators achieved stable operation up to 98% gas-fill in the best orientation and up to 80% in the worst orientation (Fig. 6b)—considerable improvements over the 90% and 70% thresholds observed with air, respectively. Fig. 6c provides a direct comparison of the force-strain curves of air-filled

and HDSG-filled actuators. As a result, the maximum specific energy increased to 51.4 J kg⁻¹ at 98% HDSG-fill in the best orientation, and 30.0 J kg⁻¹ at 95% HDSG-fill in the worst orientation (Fig. 6d).

We further evaluated the dynamic performance of HDSG-filled actuators using the same method used as in Fig. 4 (see Methods for details). As shown in Fig. 7, these actuators achieved broader bandwidths under cyclic voltage inputs in both the best and worst orientations compared with actuators having identical air-fill ratios. This improvement confirms that the use of HDSG successfully diminished the issue raised in the *Dynamic performance* section: at high actuation frequencies, dynamic splashing of the liquid dielectric reduces its local volume near the zipping front, leading to dielectric breakdown through air and lower performance. In contrast, actuator response to a step voltage input showed negligible improvement over the air-filled version (Supplementary Fig. 17).”

Reviewer 2

“The authors have addressed my previous concerns. The paper is recommended for publication.”

Response:

We appreciate the positive feedback on our revised manuscript.

Reviewer 3

Reviewer 3, general comment:

“The authors have responded to most of the questions and provided a substantial amount of additional work and experiments. In particular, the in-depth analysis of breakdown phenomena is a very valuable addition to the manuscript, enhancing both its novelty and impact. Moreover, the theoretical elaborations adequately address my previous comments. There are, however, a few aspects that could be considered in a further minor revision”

Reviewer 3, comment 1:

“Durability tests – Long-term durability tests have been conducted, achieving 100k cycles before degradation. The authors should provide a critical discussion of this result, as it appears to represent a worsening compared to the durability previously obtained with other multilayer liquid/solid actuators proposed by their group and by other research teams.”

Response:

We thank the reviewer for raising this important point, as we might not have been clear enough in describing our experimental results regarding durability of gas filled actuators. All the actuators we tested survived 100k cycles without failure; we only observed a decay of actuator strains starting at cycle numbers beyond 1k, which we attribute to charge retention effects [R1] stemming from the use of single polarity signals in these tests. Importantly, the actuator without any gas filling exhibited a very similar decay of actuation strain with frequency, indicating that the use of gas filling does not substantially change durability of actuators up to 100k cycles. Systematic testing of actuator lifetime, including proper statistics, is a very time-consuming undertaking, and beyond the scope of this paper.

To clarify the above points, we edited a paragraph in the section *Dynamic performance* in the main text.

[R1] I.-D. Sîrbu et al., *Electrostatic actuators with constant force at low power loss using matched dielectrics*, Nat. Electron., 6, 888-899, 2023

Main text, page 13, line 294: “We assessed the long-term durability of the actuators through 100,000-cycle tests. All tested actuators survived 100,000 cycles without failure; we only observed a decay of actuator strains starting at cycle numbers beyond 1,000, which we attribute to charge retention effects⁵² stemming from the use of single polarity signals in these tests. Importantly, the actuator without any gas filling exhibited a very similar decay of actuation strain over cycles, indicating that the use of gas filling does not substantially change durability of actuators up to 100,000 cycles (see Supplementary Fig. 6b and Supplementary Note).”

Reviewer 3, comment 2:

“Bandwidth testing – I am moderately skeptical about the bandwidth tests. It is difficult to explain why an increase in amplitude is observed in most cases. For example, actuators with 0% air show an amplification of almost 6 dB at 40 Hz. Could this not be attributable to a structural resonance due to the mass of the attached frame? Wouldn't this compromise the evaluation of the actual bandwidth of the actuation system?”

Response:

We thank the reviewer for this thoughtful observation regarding the amplitude increase, peaking around 40 Hz. Upon closer investigation, we found that the observed behavior originates from the internal motion of the liquid within the pouch—an intrinsic dynamic characteristic of electrohydraulic actuators—rather than from structural resonance of the test setup. We first carefully analyzed all possible resonance modes of the test setup, and concluded that all resonance modes would occur at frequencies far away from the frequency for which we observed amplitude increase (peaking at 40 Hz). We then further experimentally characterized the dynamics of the actuator, following the approach of Rothemund et al. [R1]. We measured the rise time ($t_{r,overshoot} = 0.016$ s; the time to reach the overshoot displacement) and fall time ($t_f = 0.017$ s) under 6 kV step voltage input and 100 g of external loading. These times are comparable to half the period of 30 Hz ($T/2 = 0.0167$ s), suggesting that the transient movement of the liquid becomes synchronized with the driving frequency of the actuator for frequencies around 30 Hz, amplifying displacement amplitude. This explanation is consistent with the observation that this amplification effect is reduced when air is introduced—more turbulent liquid motion disrupts the in-phase motion.

These new findings are added in the section *Dynamic performance* in the main text, and as Supplementary Figure 14.

[R1] P. Rothemund et al., *Dynamics of electrohydraulic soft actuators*, Proc. Natl. Acad. Scid. USA, 117, 16207-16213, 2020

New Supplementary Figure 14:

“

Supplementary Figure 14. Evaluation of rising and falling responses of the actuators when driven with square wave voltage signals. a Step voltage input (6 kV) lasting for 1 s and the corresponding actuator displacement, with the rising and falling responses shown in **b** and **c**, respectively. **b** Rising response of the actuator. The time to reach the overshoot displacement is $t_{r,overshoot} = 0.016$ s. **c** Falling response of the actuator when the voltage is turned off. The time to return to the initial position is $t_{falling} = 0.017$ s. These rising and falling times correspond to half the period of 30 Hz ($T/2 = 0.0167$ s).

”

Main text, page 12, line 280:

“The amplitude of the 0% air-filled reference actuator increases with frequency, peaking around 40 Hz. We found that this behavior is related to the internal motion of the liquid within the pouch—an intrinsic dynamic characteristic of electrohydraulic actuators; for frequencies around 30 Hz the motion of the liquid dielectric inside the pouch becomes synchronized with the driving frequency of the actuator, amplifying displacement; for details see Supplementary Fig. 14.”

Reviewer 3, comment 3:

“Use of C₄F₇N gas – The additional tests employing C₄F₇N gas are quite interesting. I would encourage the authors to comment on the possible toxicity of this gas, and whether they see any limitations in its use for their actuators, or conversely, if they consider it to be perfectly safe under all conditions without special precautions.”

Response:

We thank the reviewer for this useful suggestion. The safety of C₄F₇N (commercially known as Novec™ 4710, 3M™) has been well studied [R1], and detailed safety information is available in the manufacturer’s Safety Data Sheet [R2]. According to these sources, C₄F₇N is not classified as a CMR substance (carcinogenicity, mutagenicity, reproductive toxicity) and exhibits low acute inhalation toxicity (4-h LC₅₀ > 10,000 ppm, GHS classification of Category 4 or higher).

In our case, the gas is fully enclosed within the actuator pouch after fabrication, therefore when fabricated in a controlled environment—such as a well-ventilated area or fume hood—the risks after being sealed are minimal. However, caution is required if actuator breakdown occurs, as localized arcing may produce toxic decomposition byproducts (e.g., CO, CO₂, HF). Appropriate precautions should therefore be taken during disposal or in the event of electrical failure.

A brief discussion summarizing these considerations has been added to the **Discussion** section as below.

[R1] J. Owens et al., *Recent development of two alternative gases to SF₆ for high voltage electrical power applications*, *Energies*, 14(16), 5051, 2021

[R2] Safety Data Sheet, 3M™ Novec™ 4710 Insulating Gas; 3M Company: St. Paul, MN, USA, 2019

Main text, page 17, line 389:

“C₄F₇N (also known as Novec™ 4710, 3M™) exhibits low acute inhalation toxicity (4-h LC₅₀ > 10,000 ppm) and is not classified as carcinogenic, mutagenic, or reprotoxic^{56,61}. Since the gas remains sealed within the actuator pouch, normal operation poses minimal risk when fabrication is performed in a well-ventilated area or fume hood. However, electrical breakdown can generate toxic decomposition byproducts (e.g., CO, CO₂, HF), so appropriate precautions are required during disposal or in the event of actuator failure.”